# Semantic Editing with Coupled Stochastic Differential Equations

**Jianxin Zhang** [1]  **Clayton Scott** [1]

## Abstract

Editing the content of an image with a pretrained text-to-image model remains challenging. Existing methods often distort fine details or introduce unintended artifacts. We propose using *coupled stochastic differential equations* (coupled SDEs) to guide the sampling process of any pretrained generative model that can be sampled by solving an SDE, including diffusion and rectified flow models. By driving both the source image and the edited image with the same correlated noise, our approach steers new samples toward the desired semantics while preserving visual similarity to the source. The method works out-of-the-box—without retraining or auxiliary networks—and achieves high prompt fidelity along with near-pixel-level consistency. These results position coupled SDEs as a simple yet powerful tool for controlled generative AI.

## 1. Introduction

Semantic editing, as shown in Figure 1, refers to the task in which, given a source image (optionally accompanied by a prompt describing it), a target prompt, and a pretrained text-to-image model, the goal is to generate a new image that aligns with the target prompt while preserving visual similarity to the source image.

A common semantic editing pipeline is based on prominent generative models for images, which transform noise to data (Ho et al., 2020; Song & Ermon, 2021; Lipman et al., 2023; Chen et al., 2018; Liu et al., 2022). In this pipeline, inversion of the generative process maps the reference image to noise, after which the generative process is modified by conditioning on the target prompt (Meng et al., 2022;

Song et al., 2021; Mokady et al., 2022; Rout et al., 2025a). Existing implementations of this pipeline often compromise faithfulness, either because the new sampling path is independent of the one producing the source image or because the guidance relies on heuristic attention manipulations.

We propose a simple alternative based on coupled stochastic differential equations (SDEs) that can be used in conjunction with pre-trained diffusion and rectified flow models, or more generally with any model that can be sampled by solving an SDE. Our key observation builds on the time-reversal theorem of Anderson (1982), which states that the reverse dynamics of a forward SDE can be determined pathwise using a backward Brownian motion path dependent on the forward path. If the same backward noise path is reused to drive a second reverse process guided by the target prompt, then the two processes remain synchronized at the level of stochastic fluctuations and differ only through their drifts induced by the different prompts. This leads to *sync-SDE*, a plug-and-play coupling of reverse SDEs that preserves structure without retraining, optimization procedures, or auxiliary networks.

Our contributions are outlined as follows:

- We introduce *sync-SDE*, a training-free, optimization-free semantic editing method that couples reverse-time dynamics by reusing the same backward Brownian path for the reference and target processes.

- We provide a concise optimal-transport interpretation: synchronous coupling is a greedy choice for optimal bicausal Monge transports with a local quadratic cost.

- Through quantitative and qualitative evaluations, we show that sync-SDE achieves stronger prompt adherence and smaller deviations from source images than existing methods.

## 2. Related Work

Diffusion models (Ho et al., 2020; Song & Ermon, 2021) and flow-based models (Lipman et al., 2023; Chen et al., 2018; Liu et al., 2022) generate data by mapping noise to the target distribution through stochastic or ordinary differential equations. For brevity, we refer to both as *differential-equation-based generative models*. A common strategy for

[1]Electrical Engineering and Computer Science, University of Michigan, Ann Arbor, MI 48109, U.S.. Correspondence to: Jianxin Zhang <jianxinz@umich.edu>, Clayton Scott <clayscot@umich.edu>.

*Proceedings of the 43rd International Conference on Machine Learning*, Seoul, South Korea. PMLR 306, 2026. Copyright 2026 by the author(s).

| Original | Edited | Original | Edited | Original | Edited |
|---|---|---|---|---|---|

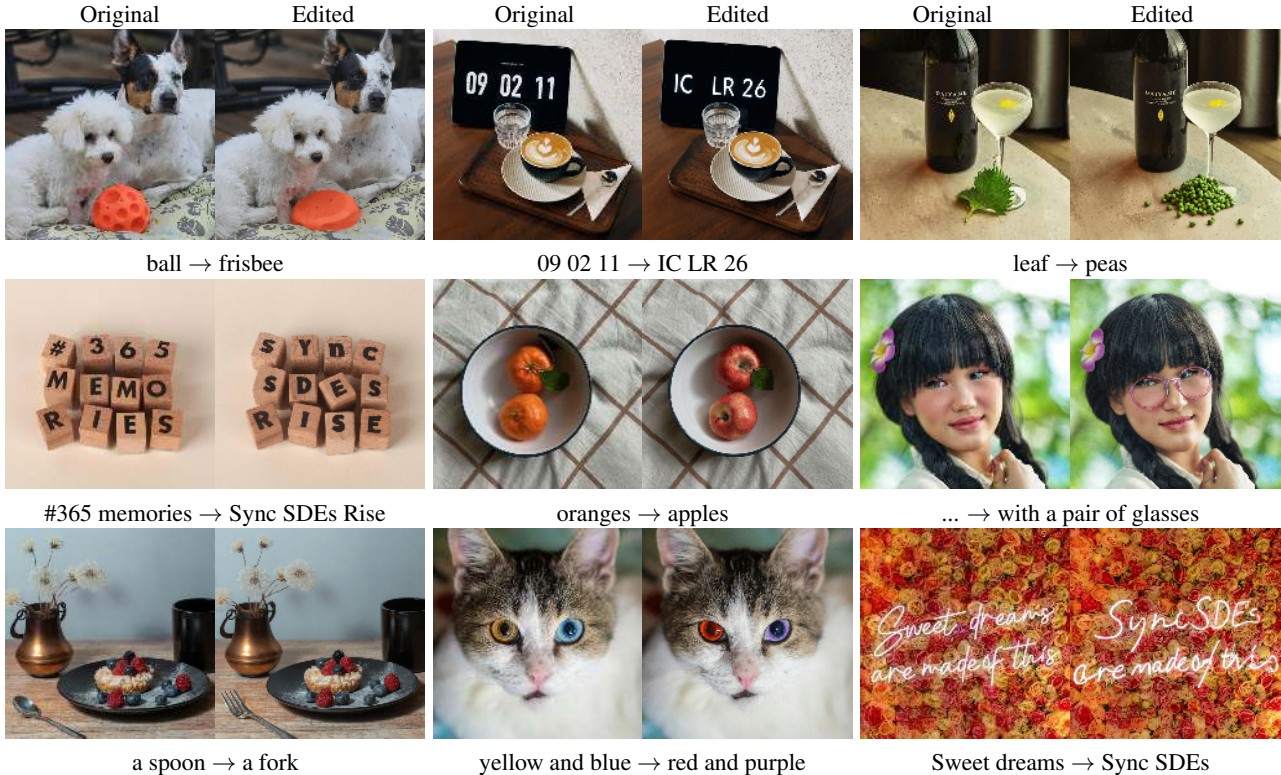

*Figure 1.* Our sync-SDE method performs text-guided image editing without retraining, test-time optimization, or model-specific modifications. By coupling the reverse-time dynamics of the source and target processes through a structured identical backward Brownian path, sync-SDE preserves fine-grained structure from the original while adapting semantics to the target prompt. Each pair shows the source image and the edited result. The text below each pair indicates the shift from the source prompt to the target prompt; full prompts are omitted due to space constraints. All edited images in this work are produced with Flux.1[dev] (Black Forest Labs, 2024).

semantic editing with such pretrained models first inverts a given image into its corresponding structured noisy representation to initialize sampling, a process known as *inversion*, and then modifies the subsequent sampling dynamics to guide the generation toward the desired semantic target, a process referred to as *editing*.

Modern large text-to-image models typically employ transformer architectures with attention blocks (Black Forest Labs, 2024; Rombach et al., 2021). Attention sharing leverages this architecture to control sampling dynamics for editing by partially or fully reusing the $(Q, K, V)$ (query, key, value) triplet from the source image when sampling for the target prompt. Hertz et al. (2023), Dalva et al. (2025), Xu et al. (2025) propose techniques to manipulate shared attention, ensuring the new sample remains visually similar to the source image. However, the experiments of Dalva et al. (2025) are limited to synthetic images, and they acknowledge challenges in editing real images due to the absence of adapted inversion procedures. Brack et al. (2024) leverage shared attention to identify local objects for editing while leaving other areas unchanged. Deng et al. (2024) incorporate a set of attention manipulation techniques into their implementation, including adding or replacing $Q, K$, or $V$

in the sampling process with those constructed from the source image.

SDEdit (Meng et al., 2022), a pioneering work for inversion, injects noise into an image, treating the result as a structured noisy representation. DDIM inversion (Song et al., 2021) is the ODE counterpart of SDEdit, where predicted noise is added to an image through an ODE. In both cases, the structured noisy representation initializes new dynamics with a different prompt to perform editing. However, both methods can lose faithfulness to the original image because the new sampling dynamics are not explicitly constrained to preserve its content. Zhao et al. (2022) use a pretrained classifier as an energy function to guide the sampling process toward the target image. Huberman-Spiegelglas et al. (2024) and Wu & la Torre (2023) explore alternative heuristics to invert and manipulate DDPM sampling (Ho et al., 2020). Chen et al. (2024) introduce a method that manipulates the noise representation obtained through DDIM inversion using a provided mask. NTI (Mokady et al., 2022) addresses low faithfulness by inverting an image via dynamic optimization of the null text prompt to match the predicted image from a structured noisy representation similar to the original. However, it tends to be less efficient due to its reliance on

test-time optimization and requires an additional attention-sharing mechanism between the source image and the new sampling process, as in Hertz et al. (2023), to ensure consistency with the target prompt. RF-inversion (Rout et al., 2025a) leverages insights from optimal control theory to design the inversion and editing processes. FlowEdit (Kulikov et al., 2024) reinterprets the inversion process, mapping the sampling ODE from the noise space back to the image space. FireFlow (Deng et al., 2024) and RF-Edit (Wang et al., 2025a) propose solvers better adapted to rectified-flow inversion and employ attention sharing between the source image and the new sampling process to perform editing. DNAEdit (Xie et al., 2025) refines model predictions within the sampling dynamics, using intermediate states from the inversion to align the generated sample with the source image. Additional methods such as (Nie et al., 2023; Mou et al., 2024) target specific editing tasks like dragging and resizing objects.

Controlled generation more broadly addresses steering generative models toward user-specified objectives or constraints, of which semantic editing is a special case. Recent studies (Rout et al., 2025a; Wang et al., 2025b; Rout et al., 2025b) have drawn connections between guided sampling in diffusion and stochastic optimal control, providing a theoretical lens for designing guidance algorithms. These works suggest that established control-theoretic tools, such as Pontryagin's maximum principle (Pontryagin, 1987) and numerical methods like EMSA (Li et al., 2017), could inform principled strategies in generative modeling. However, optimal control methods usually involve iterative optimization and are therefore far less efficient.

Concurrent work has explored several directions closely related to semantic editing. One line improves rectified-flow editing by decomposing or aggregating flows, constructing target-aware intermediate states, stabilizing inversion and velocity fields, or anchoring the editing trajectory with adaptive constraints (Yoon et al., 2025; Wang et al., 2025c; 2026; Dao et al., 2026). Another line focuses on preserving source content through feature or attention reuse, latent and attention mixing, frequency-domain interactions, or structured scene-level guidance (Yang et al., 2025; Fu & Okatani, 2026; Vo et al., 2026). A complementary direction addresses practical robustness issues such as high-resolution editing, dynamic-resolution acceleration, and stable near-reversible ODE solvers (Kim et al., 2026; Yan et al., 2026; Barancikova et al., 2026). Coupling multiple diffusion processes has also been studied for multi-view consistent editing (Alzayer et al., 2025), resonating with our use of coupled stochastic dynamics while targeting a different consistency objective. These works are complementary to *sync-SDE*. They mainly improve inversion, intermediate-state construction, attention/feature reuse, resolution/efficiency, or auxiliary guidance, whereas *sync-SDE* directly couples reverse-time

stochastic paths in a training-free, optimization-free, and largely architecture-agnostic manner.

## 3. Mathematical Preliminaries

In this section, we introduce the mathematical background for our semantic editing technique. After reviewing stochastic differential equations, we present the concept of coupled SDEs (Eberle, 2016; Bion–Nadal & Talay, 2019; Robinson & Szölgyenyi, 2024; Cont & Lim, 2024), and then discuss the time-reversal theorem for SDEs in Anderson (1982).

Let $(\Omega, \mathcal{F}, \mathbb{P})$ be a probability space. We denote by $\{X_t\}$ a random process $X : \Omega \times [0,1] \to \mathbb{R}^d$, viewed as a function mapping a sample $\omega \in \Omega$ and a timestamp $t \in [0,1]$ to a point in $\mathbb{R}^d$. The marginal at time $t$, denoted $X_t$, is the random variable $\omega \mapsto X(\omega, t)$ for a fixed $t \in [0,1]$.

### 3.1. Stochastic Differential Equations

A stochastic differential equation describes a continuous-time random process on a given time interval, here taken to be $[0,1]$. Such an equation takes the form[1]

$$dX_t = f(t, X_t)dt + g(t)dW_t,$$
$$\text{or equivalently,}$$
$$X_t = X_0 + \int_0^t f(s, X_s)ds + \int_0^t g(s)dW_s$$

where $f$ and $g$ are functions, and $\{W_t\}$ is a standard Brownian motion. Unless otherwise stated, all stochastic integrals with respect to $\{W_t\}$ are understood in the Itô sense. The equation specifies the evolution of $\{X_t\}$ from $t = 0$ to $t = 1$ in terms of its infinitesimal increments $dX_t$. Throughout this work, we refer to $\{X_t\}$ as a *forward SDE* if $X_0$ lies in the data domain and $X_1$ lies in the noise domain, and as a *reverse-time SDE* in the opposite case.

### 3.2. Coupled SDEs

Now consider two SDEs of the form

$$dY_t = f_1(t, Y_t)dt + g(t)dW_t^1,$$
$$dZ_t = f_2(t, Z_t)dt + g(t)dW_t^2,$$

where $\{W_t^1\}$ and $\{W_t^2\}$ are standard Brownian motions. If $\{W_t^1\}$ and $\{W_t^2\}$ are correlated, then so too are $\{Y_t\}$ and $\{Z_t\}$, in which case these are examples of *coupled SDEs*. The joint law of $(\{W_t^1\}, \{W_t^2\})$ influences, for each realization, the relative trajectories of $\{Y_t\}$ and $\{Z_t\}$, such as whether $Y_t$ stays close to, moves away from, or intersects $Z_t$.

---

[1]More generally, $g$ can also depend on $X_t$, but in common diffusion, $g$ depends only on $t$.

The most notable among coupling strategies are *synchronous coupling* and *reflection coupling*. In synchronous coupling, $W_t^2 = W_t^1$. In this case, the noise driving $Y_t$ is identical to that of $Z_t$. Synchronous coupling is known to minimize [2] a certain modified Wasserstein-2 distance between $Y_t$ and $Z_t$ when both processes are real-valued (Bion–Nadal & Talay, 2019; Robinson & Szölgyenyi, 2024). In reflection coupling, $dW_t^2 = (I - 2n_t n_t^T)dW_t^1$, where $n_t = \frac{Y_t - Z_t}{\|Y_t - Z_t\|}$(Eberle, 2016). This construction, introduced by Lindvall & Rogers (1986) in order to control the total variation distance of the distributions of $Y_t$ and $Z_t$ at a given time $t \in [0, 1]$, reflects the noise driving $Z_t$ along the direction of $Y_t - Z_t$.

### 3.3. Time-Reversal of SDEs

In this section, we present the time-reversal theorem from Anderson (1982) adapted to our context, which provides a precise formulation of the reverse-time SDE corresponding to a given forward SDE.

**Theorem 3.1** (Anderson (1982))**.** *Consider the forward SDE, $dX_t = f(t, X_t)dt + g(t)dW_t$, where $t \in [0, 1]$, $X_t$ takes values in $\mathbb{R}^d$, $f : [0, 1] \times \mathbb{R}^d \to \mathbb{R}^d$ and $g : [0, 1] \to \mathbb{R}$ are such as to guarantee the existence of a unique strong solution [3] and the existence of a differentiable density of the marginal distribution of $X_t$, and $\{W_t\}$ is a standard Brownian motion. Let $p_t(x)$ denote the marginal density of $X_t$. Define the $\mathbb{R}^d$-valued process*

$$\overline{W}_t = W_{1-t} - W_1 + \int_0^{1-t} g(s)\nabla_x \log p_s(X_s)ds. \quad (1)$$

*Then $\{\overline{W}_t\}_{t \in [0,1]}$ is a standard Brownian motion with respect to the reversed filtration $\{\overline{\mathcal{A}}_t\}_{t \in [0,1]}$, i.e., $\overline{\mathcal{A}}_t$ is the minimal $\sigma$-algebra that makes $\{X_s : s \in [1 - t, 1]\}$ and $\{\overline{W}_s : s \in [0, t]\}$ measurable. A reverse-time SDE for $X_t$, $t \in [0, 1]$, has the form*

$$d\overline{X}_t = \left[-f(1 - t, \overline{X}_t) + g^2(1 - t)\nabla_x \log p_{1-t}(\overline{X}_t)\right] dt$$
$$+ g(1 - t)d\overline{W}_t, \quad (2)$$

*with $\overline{X}_0 = X_1$, i.e., $\{\overline{X}_t\} = \{X_{1-t}\}$.*

Note that the equality $\{\overline{X}_t\} = \{X_{1-t}\}$ is meant in a pathwise sense. Namely, the stochastic process, viewed as a function $\overline{X} : [0, 1] \times \Omega \to \mathbb{R}^d$, satisfies $\overline{X}(t, \omega) = X(1 - t, \omega)$.

---

[2]Strictly speaking, synchronous coupling is optimal among *bicausal* couplings on $\mathbb{R}$, which intuitively restricts the coupling so that each process can only use information available from the other's past (and not its future). We omit the mathematical details here and defer the discussion of bicausal coupling to the appendix.

[3]For readers unfamiliar with the terminology, a *strong solution* is adapted to a given Brownian motion, while a *weak solution* is a pair $(Y_t, W_t)$ constructed together that formally satisfies the SDE (Øksendal, 2003). This distinction is not essential for following the rest of the paper.

This result gives a pathwise correspondence: if a forward trajectory is generated with a Brownian motion $W_t$, then integrating (2) with $\overline{W}_t$ exactly retraces the forward path in reverse time.

At first glance, $\{\overline{W}_t\}$ in (1) may look nothing like a Brownian motion. This is because Brownian motion is always defined relative to a filtration, an evolving information set. With respect to the forward filtration, $\{\overline{W}_t\}$ indeed carries "insider" knowledge of the future and thus is not Brownian. However, with respect to the reversed filtration $\{\overline{\mathcal{A}}_t\}_{t \in [0,1]}$, that future becomes the new past and the score function term in (1) simply removes the predictable drift from this insider view. In the appendix, we present an example where a process is a Brownian motion $w.r.t.$ one filtration but fails to be a Brownian motion $w.r.t.$ another filtration.

## 4. Ornstein Uhlenbeck process and SDE sampling

In the context of generative modeling, Theorem 3.1 underlies the standard SDE sampling procedure (Song & Ermon, 2021). Let $p(\cdot|c)$ be the probability of a datapoint conditioned on a variable $c$ (*e.g.*, a prompt). Let $p_t(\cdot|c)$ be the marginal density of $X_t$ when $X_0 \sim p(\cdot|c)$. Let $X_0 \sim p(\cdot|c)$ and suppose the forward process $\{X_t\}$ is an Ornstein–Uhlenbeck (OU) process,

$$dX_t = -\alpha(t)X_t dt + g(t) dW_t, \quad (3)$$

as in many popular diffusion models (Song & Ermon, 2021) and the rectified SDE (Rout et al., 2025a), an SDE formulation of rectified flow (see below). The unique strong solution of (3) admits the form

$$X_t = m(t)X_0 + \int_0^t \Phi(t, s)g(s)dW_s, \quad (4)$$

where $m(t) = \exp\left(-\int_0^t \alpha(u)du\right)$ and $\Phi(t, s) = \exp\left(-\int_s^t \alpha(u)du\right)$.

To sample from $p(\cdot \mid c)$ with a trained model, one integrates the reverse-time SDE

$$d\overline{X}_t = \left[\alpha(1 - t)\overline{X}_t + g^2(1 - t)S(\overline{X}_t, c, 1 - t)\right] dt$$
$$+ g(1 - t) d\widetilde{W}_t,$$

where $S$ approximates the score $\nabla_x \log p_t(x \mid c)$ and $\widetilde{W}_t$ is an independent Brownian motion. Theorem 3.1 ensures that $\overline{X}_1$ has the correct distribution, but with an independent Brownian motion, the generated sample need not match a specific source image pathwise.

Rout et al. (2025a) showed that the rectified flow ODE (Liu et al., 2022) shares the same marginals for all $t \in [0, 1]$

as the SDE with $\alpha(t) = \frac{1}{1-t}$ and $g(t) = \sqrt{\frac{2t}{1-t}}$. Thus, a pretrained rectified flow $d\overline{X}_t = v_\theta(\overline{X}_t, c, t)\, dt$ is described by the SDE $d\overline{X}_t = \left[2v_\theta(\overline{X}_t, c, t) + \alpha(1-t)\overline{X}_t\right]dt + g(1-t)\, d\widetilde{W}_t$ with the above $\alpha$ and $g$, where $v_\theta$ is the pretrained model with weights $\theta$. This enables sampling with Flux (Black Forest Labs, 2024), a rectified flow model, via an SDE.

## 5. Semantic Editing by sync-SDE

In the setting of semantic editing, let $y_0$ be a given source image, $c_{\text{src}}$ a prompt describing $y_0$, and $c_{\text{tar}}$ the editing prompt specifying the desired output. Let $S(y_t, c, t)$ denote a pretrained neural network that approximates the score function $\nabla_x \log p_t(y_t \mid c)$, taking as input a state $y_t$, a conditioning prompt $c$, and a time $t \in [0, 1]$. For flow-based models that do not directly parameterize $\nabla_x \log p_t(y_t \mid c)$, the learned quantity can be converted into a score approximation through simple algebraic transformations (Rout et al., 2025a). Our objective is to modify the reverse-time dynamics so that the generated image is consistent with the target prompt $c_{\text{tar}}$ while preserving visual similarity to the source image $y_0$.

We now apply the ideas of coupled SDEs to semantic image editing, establishing a conceptually independent framework *w.r.t.* prior approaches. To formalize this, let $\{Y_t\}$ and $\{Z_t\}$ be solutions of forward SDEs corresponding to the reference and edited images. Consider the forward SDEs of the source and target images and the reverse-time SDEs to couple,

$$dY_t = -\alpha(t)Y_t\, dt + g(t)\, dW_t^Y, \tag{5}$$

$$dZ_t = -\alpha(t)Z_t\, dt + g(t)\, dW_t^Z, \tag{6}$$

$$d\overline{Y}_t = \alpha_1^*(Y_t)dt + g(1-t)\, d\overline{W}_t^Y, \tag{7}$$

$$d\overline{Z}_t = \alpha_2^*(Z_t)dt + g(1-t)\, d\overline{W}_t^Z, \tag{8}$$

where

$$\alpha_1^*(\overline{Y}_t) = \alpha(1-t)\overline{Y}_t + g^2(1-t)\nabla_x \log p_{1-t}(\overline{Y}_t \mid c_{\text{src}}),$$
$$\alpha_2^*(\overline{Z}_t) = \alpha(1-t)\overline{Z}_t + g^2(1-t)\nabla_x \log p_{1-t}(\overline{Z}_t \mid c_{\text{tar}}).$$

Given a source image $y_0$, the edited image is simulated as follows: first, sample $(w_t) \sim W_t^Y$, with $W_t^Y$ being an independent Brownian motion, to evolve $y_0$ toward a noisy image $y_1$ via (5). Next, simulate the Brownian motion path $(\overline{w}_t)$ using (1) with realizations $(w_t)$ and $(y_t)$ and using prompt $c_{\text{src}}$, so that $(\overline{w}_t)$ is a realization of $\overline{W}_t^Y$. Finally, simulate $\overline{z}_1 = z_0$ with (8), initializing at $\overline{z}_0 = y_1$, driven by the Brownian path $(\overline{w}_t)$ with prompt $c_{\text{tar}}$. The central idea is to use a shared Brownian motion $\{\overline{W}_t^Z\} = \{\overline{W}_t^Y\}$ between (8) and (7), making the two SDEs synchronously coupled. An implementable description of this procedure is presented in Algorithm 1. Note that in Algorithm 1, we assume the time grid is symmetric for ease of presentation, *i.e.*, $1 - t_k \in \{t_k\}_{k=1}^N, \forall k = 0, \ldots, N$; this is not required in practice.

---

**Algorithm 1** sync-SDE Semantic Editing

---

**Require:** Source image $y_0$, source prompt $c_{\text{src}}$, target prompt $c_{\text{tar}}$, score network $S(\cdot, \cdot, \cdot)$, symmetric time grid $0 = t_0 < \cdots < t_N = 1$, $\alpha$ and $g$ defining the OU process.

1: Sample $\{\Delta W_{t_k}\}_{k=0}^{N-1}$ with $\Delta W_{t_k} \overset{i.i.d.}{\sim} \mathcal{N}(0, \Delta t_k I_d)$ and $\Delta t_k = t_{k+1} - t_k$.

2: For $k = 0, \ldots, N$, compute the forward path with (4):

$$Y_{t_k} \leftarrow m(t_k)y_0 + \sum_{j=0}^k \Phi(t_k, t_j)g(t_j)\Delta W_{t_j}.$$

3: Define reversed path $\overline{Y}_{t_k} \leftarrow Y_{1-t_k}$ for $k = 0, \ldots, N$.

4: For $k = 0, \ldots, N$, construct structured backward Brownian increments with (1):

$$\Delta \overline{W}_{t_k} \leftarrow -\Delta W_{t_k} - g(1-t_k)S\left(\overline{Y}_{t_k}, c_{\text{src}}, 1-t_k\right)\Delta t_k.$$

5: Initialize $\overline{Z}_0 \leftarrow Y_{t_N}$.

6: **for** $k = 0$ to $N - 1$ **do**

7: $\quad b_Z \leftarrow \alpha(1-t_k)\overline{Z}_{t_k} + g^2(1-t_k)S\left(\overline{Z}_{t_k}, c_{\text{tar}}, 1-t_k\right).$

8: $\quad \overline{Z}_{t_{k+1}} \leftarrow \overline{Z}_{t_k} + b_Z \Delta t_k + g(1-t_k)\Delta \overline{W}_{t_k}.$

9: **end for**

10: **return** Edited image $\overline{Z}_{t_N}$.

---

## 6. An optimal transport interpretation

Coupling the reverse-time dynamics can be interpreted through the lens of bicausal optimal transport. Consider the reference and target SDEs with potentially correlated Brownian motions $(\overline{W}_t^Y, \overline{W}_t^Z)$ in (7) and (8), respectively. Let $\overline{\mathbb{Y}}$ and $\overline{\mathbb{Z}}$ denote their laws, *i.e.*, the probability distribution on their sampled paths, respectively. By Theorem 3.4 of Cont & Lim (2024), the optimal bicausal Monge transport[4] between $\overline{\mathbb{Y}}$ and $\overline{\mathbb{Z}}$ can be written in the form

$$d\overline{Z}_t = [\alpha(1-t)\overline{Z}_t + g^2(1-t)\nabla_x \log p_{1-t}(\overline{Z}_t \mid c_{\text{tar}})]\, dt + g(1-t)Q_t d\overline{W}_t^Y,$$

*i.e.*, $d\overline{W}_t^Z = Q_t d\overline{W}_t^Y$ where $Q_t$ is an adapted orthonormal matrix process.

---

[4]Intuitively, a bicausal Monge transport is a function that transforms one diffusion path into another using only past information from both paths. We discuss the formal definition and its properties in Appendix.

This shows that designing a bicausal Monge transport between two SDEs reduces to designing an orthonormal matrix process $Q_t$. Unfortunately, finding an optimal $Q_t$ for a given transport cost in the context of semantic editing is computationally expensive, as it essentially amounts to solving an optimal control problem, where the exact solution requires backpropagating through the SDE path and incurs a substantial memory footprint (Wang et al., 2025b; Pontryagin, 1987; Li et al., 2017). This computationally heavy search for an optimal $Q_t$ runs counter to the goal of this work—an efficient method for accurate semantic editing without additional optimization or retraining—so we leave it to future work.

Synchronous coupling corresponds to $Q_t = I_d$, while reflection coupling corresponds to $Q_t = I_d - 2n_t n_t^\top$ with $n_t = (\overline{Y}_t - \overline{Z}_t)/\|\overline{Y}_t - \overline{Z}_t\|$. To see how sync-SDE is a greedy choice of bicausal Monge transport under the local quadratic cost, fix $t$ and a small step $\Delta t$. The one-step difference between the target and reference processes is

$$
\begin{aligned}
\overline{Z}_{t+\Delta t} - \overline{Y}_{t+\Delta t} &\approx \overline{Z}_t - \overline{Y}_t \\
&+ \big[b_Z(t, \overline{Z}_t) - b_Y(t, \overline{Y}_t)\big]\Delta t \\
&+ g(1-t)(Q_t - I_d)\Delta \overline{W}_t,
\end{aligned}
\tag{9}
$$

with $\Delta \overline{W}_t \sim \mathcal{N}(0, \Delta t I_d)$ and

$$
b_Y(t, x) = \alpha(1-t)x + g^2(1-t)\nabla_x \log p_{1-t}(x \mid c_{\mathrm{src}}),
$$
$$
b_Z(t, x) = \alpha(1-t)x + g^2(1-t)\nabla_x \log p_{1-t}(x \mid c_{\mathrm{tar}}),
$$

where the approximation in (9) is exact when $\Delta t \to 0$.

Conditioning on $\overline{Z}_t$ and $\overline{Y}_t$, the expected increment is

$$
\begin{aligned}
&\mathbb{E}\Big[\big\|\overline{Z}_{t+\Delta t} - \overline{Y}_{t+\Delta t}\big\|^2 \mid \overline{Y}_t, \overline{Z}_t\Big] \\
&= F(\overline{Y}_t, \overline{Z}_t) + 2g^2(1-t)\big(d - \mathrm{tr}(Q_t)\big)\Delta t,
\end{aligned}
\tag{10}
$$

where $F$ is a function depending only on $\overline{Y}_t$ and $\overline{Z}_t$, and thus is constant under the conditioning. Among all orthonormal $Q_t$, the trace is maximized by $Q_t = I_d$, so the myopic minimizer of this local quadratic deviation is the synchronous choice. We postpone the derivation to section A.3.

Due to Theorem 3.1, $\overline{\mathbb{Y}}$ and $\overline{\mathbb{Z}}$ also correspond to the law of (5) with $Y_0 \sim p(\cdot \mid c_{\mathrm{src}})$ and the law of (6) with $Z_0 \sim p(\cdot \mid c_{\mathrm{tar}})$, respectively. Thus, Sync-SDE can be viewed as a greedy transport that maps the law of (5) with $Y_0 \sim p(\cdot \mid c_{\mathrm{src}})$ to the law of (6) with $Z_0 \sim p(\cdot \mid c_{\mathrm{tar}})$. This transport is only valid for typical samples from $p(\cdot \mid c_{\mathrm{src}})$. If $c_{\mathrm{src}}$ does not describe the source image, *i.e.*, $y_0$ lies outside the support of $p(\cdot \mid c_{\mathrm{src}})$, the transport provides no guarantee for that case. Conversely, choosing $c_{\mathrm{src}}$ such that $y_0$ is likely under $p(\cdot \mid c_{\mathrm{src}})$ ensures the transport is well-defined and, intuitively, carries mass from high-probability regions of

the source distribution to high-probability regions of the target distribution. We confirm this intuition through the qualitative studies presented in Appendix D.3.

## 7. Experiment

In our experiments, we use the official pretrained weights of Flux.1[dev] (Black Forest Labs, 2024) from Hugging-Face. Sampling is performed using the SDE equivalence of rectified flow presented in Lemma A.4. of Rout et al. (2025a). For all methods, we fix the total number of sampling steps to 28. Since modern image generative models are typically trained on Internet-crawled data, we construct a dataset of 306 $(y_0, c_{\mathrm{src}}, c_{\mathrm{tar}})$ triplets using 91 $1024 \times 1024$ images from `pexels.com` uploaded after the release of Flux.1[dev]. The source prompts are generated with BLIP (Li et al., 2022) and refined by us, while the target prompts are modifications (by us) of the source prompts.

We compare sync-SDE with the following recent semantic editing methods built on pretrained text-to-image generative models: SDEdit (Meng et al., 2022), FlowEdit (Kulikov et al., 2024), FireFlow (Deng et al., 2024), RF-Inv (Rout et al., 2025a), and RF-Edit (Wang et al., 2025a). All except SDEdit are positioned as state-of-the-art. For all baselines, we use hyperparameters as recommended in their respective papers, codebases, or GitHub releases. Our method initiates the coupling process at $t_0 = 1/7$ rather than 0 to ensure numerical stability. In preliminary experiments, we observed that smaller $t_0$ values make structural changes easier, while larger values better preserve similarity to the source image. This choice of $t_0$ is consistent with other baseline methods, such as FlowEdit, and tends to work well across most images. All methods run in comparable time, taking 15–25 seconds on an H100 GPU for a single $1024 \times 1024$ image.

Quantitatively, we evaluate our method and competing approaches using two measures of visual change, L1 distance and LPIPS (Zhang et al., 2018) distance, alongside the CLIP (Radford et al., 2021) score between the edited images and their corresponding target prompts. Each point in Figure 2 corresponds to a specific hyperparameter setting recommended in the respective paper or official GitHub release of that method. The three points shown for sync-SDE in Figure 2 correspond to different guidance strengths when calling the Flux model with $c_{\mathrm{src}}$ and $c_{\mathrm{tar}}$, set to $\{1.0, 1.5, 2.5\}$. The plots show each distance metric (x-axis) against the corresponding CLIP score (y-axis), illustrating the trade-off between preserving source-image fidelity and achieving prompt adherence. Across all metrics, our method consistently attains higher CLIP scores while incurring smaller edits to the source image, indicating that it produces semantically aligned results with a lower "editing budget."

Qualitatively, we present the original images and their ed-

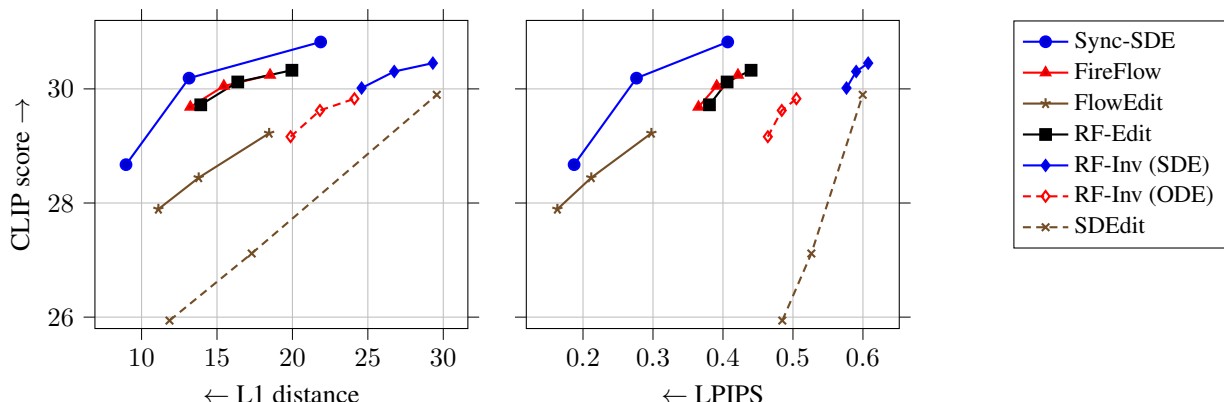

*Figure 2.* Trade-off between semantic alignment and perceptual similarity for different image editing methods. The x-axis reports distance metrics (L1 and LPIPS here), while the y-axis shows CLIP score. Points represent results for each method at different hyperparameter settings, and lines connect results from lower to higher distance. A higher CLIP score indicates better semantic consistency with the target prompt, while a lower distance means higher visual fidelity to the source image. Methods toward the upper-left corner achieve a better balance between preserving image structure and matching the edit prompt.

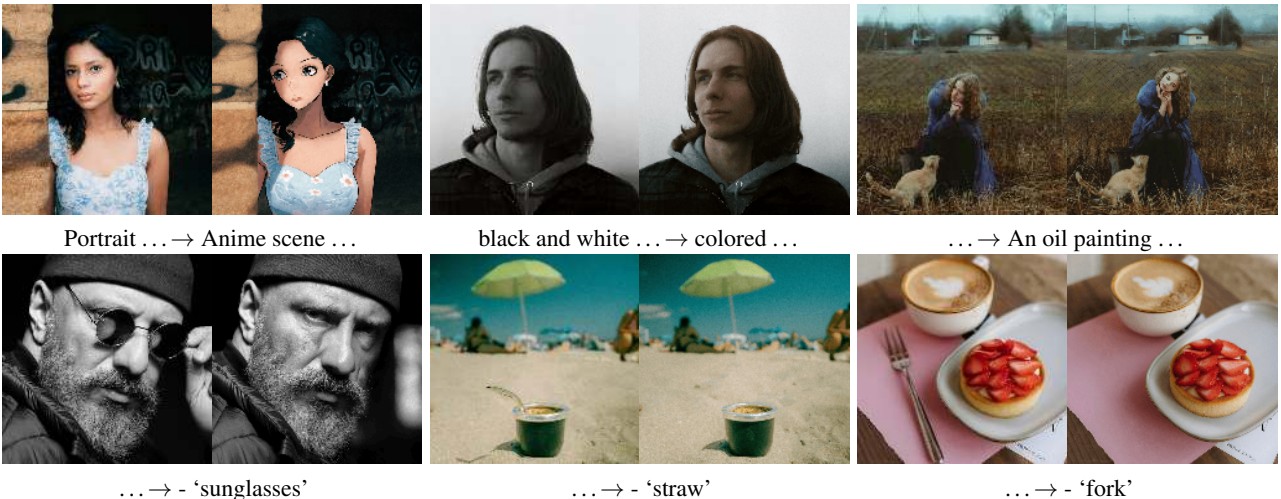

*Figure 3.* Global style transfer (the first row) and negative prompts (the second row) with sync-SDE. In each pair, the source image is shown on the left and the edited image on the right.

its produced by sync-SDE in Figure 1, and compare our method with competing approaches in Figure 4, which also includes pixel-wise difference maps, obtained by plotting the absolute pixel-wise difference between the edited and source images. In the difference maps, good edits appear as bright pixels confined to regions relevant to the target prompt, while the rest remains dark. For each example and for each method compared, we select, among the three hyperparameter settings reported in the quantitative results, the image that is most similar to the source while still showing a meaningful edit, to rule out degenerate cases where the result remains identical to the source. Sync-SDE produces edits that align closely with the target prompt while preserving the rest of the image, yielding more localized and faithful modifications than competing methods. Notably, in the first task (adding coffee), our method is the only one

that preserves the texture on the saucer. In the second task (Greek marble sculpture), competing methods often distort the material qualities and lighting of the marble, modify the head covering, or introduce unnatural features such as an Adam's apple, whereas sync-SDE alters only the facial expression as intended while faithfully preserving the marble texture and lighting. In the third task (a spoon on the table), all other methods either alter the global lighting or modify unrelated objects such as the fruits, cake, mug, wall, or dandelions. In the fourth task (adding glasses), every other method changes the person's appearance, whereas ours even preserves the eye color. In addition, sync-SDE demonstrates strong capacity for global style transfer and handling negative prompts, as shown in Figure 3.

# 8. Conclusion

We have introduced sync-SDE, a simple and efficient framework for text-guided semantic image editing that couples reverse-time SDEs through a shared backward Brownian path. Both qualitative and quantitative experimental results demonstrate that sync-SDE achieves high prompt fidelity with minimal unintended alterations, outperforming recent state-of-the-art editing methods. In Section B, we introduce *resampling-ODE*, a more stable, less hyperparameter-sensitive variant, though less effective at generating fine-grained details compared to sync-SDE.

## Impact Statement

This paper presents work whose goal is to advance machine learning methods for controllable generative image editing. Such tools may support creative and assistive image-editing applications, but they could also be misused for deceptive or unauthorized image manipulation. We encourage responsible use with appropriate consent, disclosure, and safeguards against harmful or misleading content.

Original  Sync-SDE  Fireflow  Flowedit  RF-Edit  RF-Inv (SDE)  SDEdit

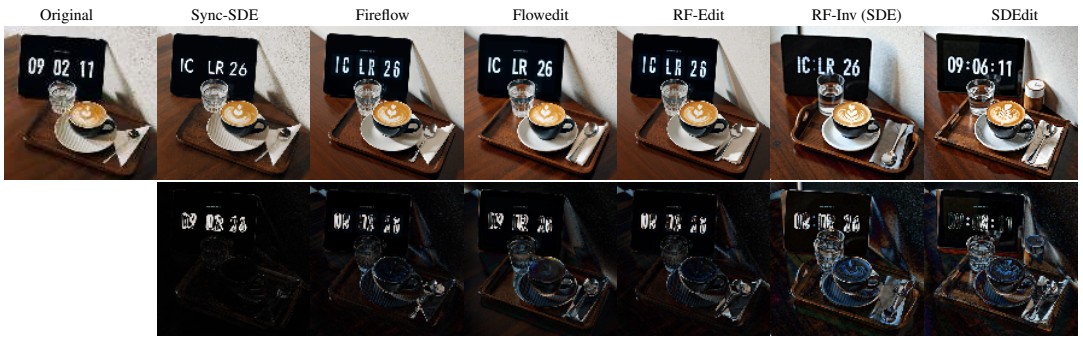

$c_{\text{src}}$ =*A latte with latte art in a black cup on a saucer, served with a glass of water and a spoon on a wooden tray, next to a digital clock display reading "09 02 11".*

$c_{\text{tar}}$ =*A latte with latte art in a black cup on a saucer, served with a glass of water and a spoon on a wooden tray, next to a digital clock display reading "IC LR 26".*

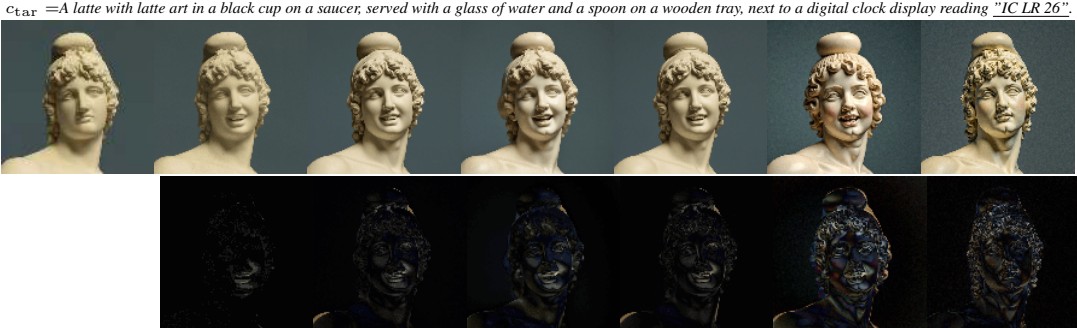

$c_{\text{src}}$ =*Close-up photograph of a classical marble bust sculpture against a plain muted blue background. The statue depicts a youthful figure with curly hair, serene facial expression, and a rounded head covering resembling a cap. The polished white marble surface shows fine detailing in the hair and smooth texture of the face, lit with soft diffused light.*

$c_{\text{tar}}$ =*Close-up photograph of . . . . The statue depicts a youthful figure, laughing happily, with curly hair and a rounded head covering resembling a cap. The polished white marble . . . .*

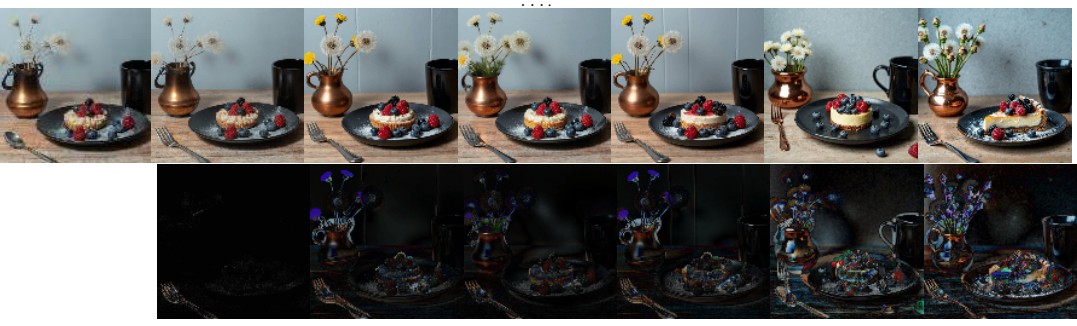

$c_{\text{src}}$ =*A dessert with berries and blueberries on a plate next to a black cup, a copper vase of dandelions, and a spoon, on the side of a wooden table with a gray wall.*

$c_{\text{tar}}$ =*A dessert with berries and blueberries on a plate next to a black cup, a copper vase of dandelions, and a fork, on the side of a wooden table with a gray wall.*

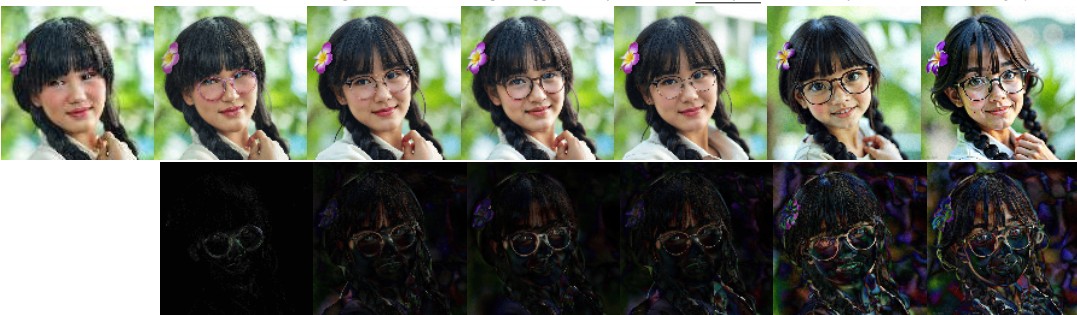

$c_{\text{src}}$ =*A close-up portrait of a woman with long dark braided hair, wearing a white top, a purple and yellow plumeria flower tucked in her hair.*

$c_{\text{tar}}$ =*A close-up portrait of a woman with a pair of glasses and long dark braided hair, wearing a white top, a purple and yellow plumeria flower tucked in her hair.*

*Figure 4.* Qualitative comparison of Sync-SDE with recent semantic editing baselines: FireFlow (Deng et al., 2024), FlowEdit (Kulikov et al., 2024), RF-Edit (Wang et al., 2025a), RF-Inv (Rout et al., 2025a), and SDEdit (Meng et al., 2022). For each image, we show the original image followed by the edited results from each method. The next row shows the corresponding pixel-wise difference maps, where brighter regions indicate larger changes.

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

# A. Mathematical Background

Let $(\Omega, \mathcal{F}, \mathbb{P})$ be a probability space. For a set $A$, let $\sigma(A)$ denote the smallest $\sigma$-algebra containing $A$. With a slight abuse of notation, for a random variable $X$, $\sigma(X)$ denotes the smallest $\sigma$-algebra with respect to which $X$ is measurable, and $\sigma(X_s : s \leq t)$ denotes the smallest $\sigma$-algebra with respect to which all $\{X_s : s \leq t\}$ are measurable. In this section, we first present an example where a process is a Brownian motion $w.r.t.$ one filtration but fails to be a Brownian motion $w.r.t.$ another, and then review the mathematical background of Bicausal Monge Transport.

## A.1. Brownian Motion

Recall that we work with a probability space $(\Omega, \mathcal{F}, \mathbb{P})$. We first define a filtration and the standard definition of a Brownian motion:

**Definition A.1** (Filtration). A *filtration* $\{\mathcal{F}_t\}_{t \geq 0}$ is an increasing family of $\sigma$-algebras, *i.e.*, $\mathcal{F}_s \subset \mathcal{F}_t \subset \mathcal{F}$, $0 \leq \forall s \leq t < \infty$.

**Definition A.2** (Brownian motion w.r.t. a filtration). A process $\{B_t\}_{t \geq 0}$ is called a standard *Brownian motion* with respect to the filtration $\{\mathcal{F}_t\}$ if:

1. $B_0 = 0$ almost surely and the sample paths are continuous;

2. $\forall t \geq 0$, $B_t$ is $\mathcal{F}_t$ measurable.

3. For $0 \leq s < t$, the increment $B_t - B_s$ is independent of $\mathcal{F}_s$ and $(B_t - B_s) \mid \mathcal{F}_s \sim \mathcal{N}(0, (t-s)I_d)$.

Here, a filtration $\{\mathcal{F}_t\}$ is simply a mathematical way of encoding the information available up to time $t$. A crucial point is that the Brownian property is defined relative to a specific filtration. Enlarging the filtration by including extra information can break the independence condition in item 3. We then see how a process could fail to be so under a different filtration.

**Example.** Let $\{B_t\}_{t \in [0,1]}$ be a standard Brownian motion with its natural filtration

$$\mathcal{F}_t = \sigma(B_s : 0 \leq s \leq t), \quad t \in [0,1].$$

Now define an enlarged filtration

$$\mathcal{G}_t = \sigma(\mathcal{F}_t \cup \sigma(B_1)), \quad t \in [0,1],$$

where $\sigma(B_1)$ is the $\sigma$-algebra generated by the terminal value $B_1$.

- $\{\mathcal{F}_t\}$**-view:** By construction, $\{B_t\}$ is a Brownian motion with respect to $\{\mathcal{F}_t\}$.

- $\{\mathcal{G}_t\}$**-view:** For $s < t < 1$, since $B_1$ is $\mathcal{G}_s$-measurable, the conditional expectation is

$$\mathbb{E}[B_1 - B_s \mid \mathcal{G}_s] = B_1 - B_s,$$

which is not almost surely zero. Thus, given $\mathcal{G}_s$, the increment is not independent of $\mathcal{G}_s$ and has a nonzero mean, hence is not a $\{\mathcal{G}_t\}$-Brownian motion.

Intuitively, under $\{\mathcal{G}_t\}$, the process is seen by an insider who knows $B_1$ in advance.

## A.2. Bicausal Monge Transport

We now give a self-contained, precise formulation of *Bicausal Monge Transport*, following the framework of Cont & Lim (2024) with adaptations to our notation.

**Path Space and filtration** Fix a positive integer $d$. Let $\mathcal{W}^d := C([0,1], \mathbb{R}^d)$ be the space of continuous paths with the supremum norm $\|\cdot\|_\infty$ and induced Borel $\sigma$-algebra $\mathcal{F}_1$. For $t \in [0,1]$, define the truncation map $f_t : \mathcal{W}^d \to \mathcal{W}^d$ by $f_t(\omega) := \omega(\cdot \wedge t)$ and the canonical filtration $\mathcal{F}_t := \sigma(f_t) \subseteq \mathcal{F}_1$. Its right-continuous version is $\mathcal{H}_t := \bigcap_{\epsilon > 0} \mathcal{F}_{(t+\epsilon) \wedge 1}$. The canonical process is $X : \mathcal{W}^d \times [0,1] \to \mathbb{R}^d$, $X_t(\omega) := \omega(t)$; we write $X_\cdot(\omega) = \omega$ for the identity on paths.

**Pushforwards** If $(A_i, \mathcal{A}_i)$ are measurable spaces, $\mathcal{A}_1 \otimes \mathcal{A}_2$ denotes the product $\sigma$-algebra on $A_1 \times A_2$. For a probability measure $\eta$ on $(A_1, \mathcal{A}_1)$, $\mathcal{A}_1^\eta$ denotes the $\eta$-completion of $\mathcal{A}_1$. For a measurable map $T : A_1 \to A_2$, its *pushforward* of $\eta$ is $T_\#\eta(A) := \eta(T^{-1}(A))$ for a $\eta$-measurable set $A$.

**Couplings, causality, and bicausality** Given $\eta, \nu \in \mathcal{P}(\mathcal{W}^d)$, a probability measure $\pi \in \mathcal{P}(\mathcal{W}^d \times \mathcal{W}^d)$ is a *coupling* of $(\eta, \nu)$ if

$$P_\#\pi = \eta \quad \text{and} \quad P'_\#\pi = \nu,$$

where $P$ and $P'$ are the first and second marginals, respectively, defined as $P(\omega, \omega') := \omega$ and $P'(\omega, \omega') := \omega'$. The set of all couplings is $\Pi(\eta, \nu)$. Since $\mathcal{W}^d \times \mathcal{W}^d$ is Polish, any $\pi \in \Pi(\eta, \nu)$ admits an $\eta$-a.s. unique probability kernel (regular conditional distribution)

$$\Theta_\pi : \mathcal{W}^d \times \mathcal{F}_1 \to [0,1], \qquad (\omega, B) \mapsto \Theta_\pi^\omega(B),$$

such that for all $A, B \in \mathcal{F}_1$,

$$\pi(A \times B) = \int_{\mathcal{W}^d} \mathbf{1}_A(\omega) \, \Theta_\pi^\omega(B) \, \eta(d\omega).$$

Heuristically, $\Theta_\pi^\omega$ is the conditional distribution of $Y$ conditioned on $X = \omega$.

**Definition A.3** (Causal, bicausal, and Monge couplings).
Let $\eta, \nu \in \mathcal{P}(\mathcal{W}^d)$ and $\Pi(\eta, \nu)$ as above.

1. **Causal coupling.** A coupling $\pi \in \Pi(\eta, \nu)$ is *causal*, from $X$ to $Y$, if for every $t \in [0, 1]$ and every $B \in \mathcal{H}_t$ the map
$$\omega \longmapsto \Theta_\pi^\omega(B)$$
is $\mathcal{H}_t^\eta$-measurable. We denote the set of causal couplings by $\Pi_c(\eta, \nu)$.

2. **Bicausal coupling.** Let $R : \mathcal{W}^d \times \mathcal{W}^d \to \mathcal{W}^d \times \mathcal{W}^d$ be the coordinate swap, $R(\omega, \omega') = (\omega', \omega)$. A coupling $\pi \in \Pi(\eta, \nu)$ is *bicausal* if
$$\pi \in \Pi_c(\eta, \nu) \quad \text{and} \quad R_\# \pi \in \Pi_c(\nu, \eta).$$
The set of all bicausal couplings is $\Pi_{bc}(\eta, \nu)$.

3. **Bicausal Monge coupling.** A *bicausal Monge coupling* is a deterministic plan $\pi_T := (X., T)_\# \eta$ induced by a measurable map $T : \mathcal{W}^d \to \mathcal{W}^d$ with $T_\# \eta = \nu$ such that $\pi_T \in \Pi_{bc}(\eta, \nu)$.

Causality means "no peeking into the future": under $\pi$, the conditional law of the $Y$-path up to time $t$ given $X$ depends only on the $X$-path up to $t$. Bicausality enforces this in both directions (also for $X$ given $Y$). A bicausal Monge coupling is the deterministic, pathwise version of this idea.

### A.3. Derivation of the Expected Increment

(10) follows from

$$\mathbb{E}\left[\|(Q_t - I_d)\Delta\overline{W}_t\|^2\right]$$
$$=\mathbb{E}\left[\mathrm{tr}(\Delta\overline{W}_t^T(Q_t - I_d)^T(Q_t - I_d)\Delta\overline{W}_t)\right]$$
$$=\mathbb{E}\left[\mathrm{tr}((Q_t - I_d)\Delta\overline{W}_t\Delta\overline{W}_t^T(Q_t - I_d)^T)\right]$$
$$=\mathrm{tr}((Q_t - I_d)\Delta t I_d(Q_t - I_d)^T)$$
$$=\mathrm{tr}((Q_t - I_d)(Q_t - I_d)^T)\Delta t$$
$$=\mathrm{tr}(Q_t Q_t^T - 2Q_t + I_d)\Delta t$$
$$=2\,\mathrm{tr}(I_d - Q_t)\,\Delta t.$$

## B. Resampling ODE

We now describe a variant, called *resampling-ODE*, of sync-SDE that removes the Brownian motion term from the target update while keeping it synchronized with a reference obtained from the forward model. Empirically, resampling-ODE is empirically more stable and less sensitive to hyperparameters, at a cost of being less effective at generating fine-grained details compared to sync-SDE. Recall the reverse-time drifts

$$b_Y(t, x) = \alpha(1 - t)x + g^2(1 - t)S(x, c_{\mathrm{src}}, 1 - t),$$
$$b_Z(t, x) = \alpha(1 - t)x + g^2(1 - t)S(x, c_{\mathrm{tar}}, 1 - t).$$

We present the resampling-ODE algorithm in Algorithm 2. This algorithm can be interpreted as evolving the difference process $D_t := \overline{Z}_t - \overline{Y}_t$ rather than simulating the full target process with an explicit Brownian motion term. By maintaining $D_t$ separately, we avoid explicitly integrating the stochastic term $g(1 - t)d\overline{W}_t$ in the target process. At each iteration, we re-simulate the reference state $\overline{Y}_t$ from the forward closed form (4) using a fresh Brownian motion path and the initial state $y_0$. This gives a new realization of the reference path that is consistent with the forward dynamics starting from the same source image. The target state is then reconstructed as $\overline{Z}_t = D_t + \overline{Y}_t$, which is equivalent to resampling $\overline{Z}_t$ conditioned on the current reference $\overline{Y}_t$ and the maintained difference $D_t$. Finally, $D_t$ is updated deterministically using the drift difference $b_Y - b_Z$, ensuring that all stochasticity in the target process comes indirectly from the re-simulated reference rather than from integrating its own Brownian increments. Like in Algorithm 1, we assume a symmetric time grid in Algorithm 2 for ease of presentation, and this is not required in practice. We show qualitative comparisons between sync-SDE and resampling-ODE in Figure 5. As shown in the quantitative results in Figure 6, resampling ODE performs reasonably well, though it is not the strongest in the L1 vs. CLIP trade-off. However, it shows clear advantages against all competing methods on the LPIPS vs. CLIP plot, where it preserves perceptual similarity to the source image better than most competing methods, highlighting its robustness in maintaining structural fidelity.

---

**Algorithm 2** resampling-ODE Semantic Editing

---

**Require:** Source image $y_0$, source prompt $c_{\mathrm{src}}$, target prompt $c_{\mathrm{tar}}$, score network $S(\cdot, \cdot, \cdot)$, symmetric time grid $0 = t_0 < \cdots < t_N = 1$

1: Initialize $D_{t_0} = 0$
2: **for** $k = 0$ to $N - 1$ **do**
3:     Sample fresh forward Brownian increments $\{\Delta W_{t_j}^{(k)}\}_{j=0}^{N-1}$ with $\Delta W_{t_j}^{(k)} \sim \mathcal{N}(0, \Delta t_j I_d)$
4:     Compute the forward path with (4): $Y_{t_k} \leftarrow m(t_k)y_0 + \sum_{j=0}^{k} \Phi(t_k, t_j)g(t_j)\Delta W_{t_j}$
5:     Set the corresponding reversed reference state $\overline{Y}_{t_k}^{(k)} \leftarrow Y_{1-t_k}^{(k)}$
6:     Reconstruct $\overline{Z}_{t_k}^{(k)} \leftarrow D_{t_k} + \overline{Y}_{t_k}^{(k)}$
7:     Compute the drifts $b_Y(t, \overline{Y}_{t_k}^{(k)})$, $b_Z(t, \overline{Z}_{t_k}^{(k)})$
8:     Update the difference $D_{t_{k+1}} \leftarrow D_{t_k} + \left[b_Y(t, \overline{Z}_{t_k}^{(k)}) - b_Z(t, \overline{Y}_{t_k}^{(k)})\right]\Delta t_k$
9: **end for**
10: **return** Reconstructed image $D_{t_N} + y_0$

---

The idea of resampling-ODE extends beyond sync-SDE and can be applied to any pair of processes $(X_t, Y_t)$ where one aims to simulate $X_1$ from $X_0$ and $Y_t$ admits a closed-

Source image    Sync-SDE    Resampling-ODE    Source image    Sync-SDE    Resampling-ODE

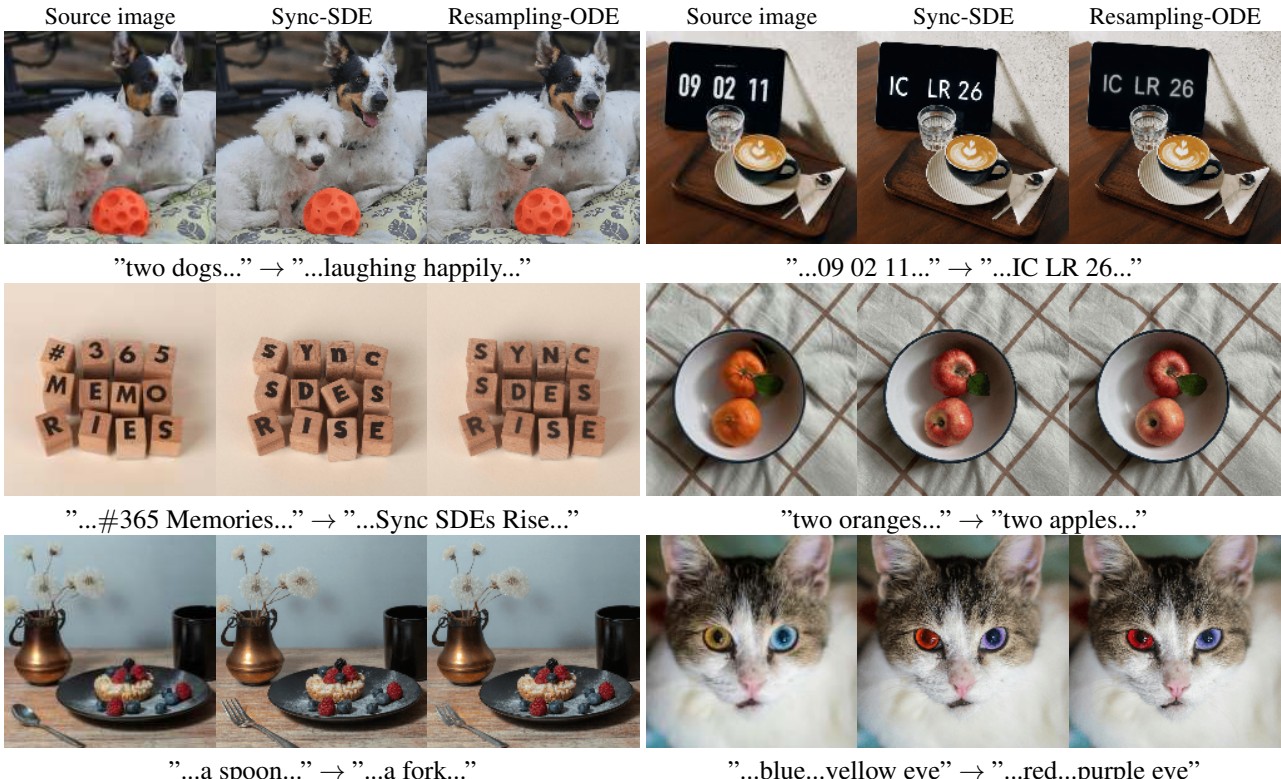

"two dogs..." → "...laughing happily..."        "...09 02 11..." → "...IC LR 26..."

"...#365 Memories..." → "...Sync SDEs Rise..."        "two oranges..." → "two apples..."

"...a spoon..." → "...a fork..."        "...blue...yellow eye" → "...red...purple eye"

*Figure 5.* Head-to-head comparison of sync-SDE and resampling-ODE methods, both producing high-quality edited images. All examples were generated with Flux.1[dev] (Black Forest Labs, 2024).

form expression. At any time $t$, we track the difference $Z_t = X_t - Y_t$, then resample a fresh $Y'_t$ and construct a copy of $X_t$ as $Z_t + Y'_t$. This mechanism resembles strategies in diffusion and flow implementations, where $X_{t+\Delta t}$ is obtained via $E[X_1 \mid X_t]$ plus freshly injected noise. While not entirely novel, this perspective highlights resampling as a general way to exploit easy-to-sample reference processes. Practically, we find it yields more stable sampling, reducing the risk of failed edits in semantic applications.

## C. Hyperparameter Choices Across Methods

For fair comparison, we evaluate three representative settings for each method, as recommended in their respective papers, codebases, or GitHub releases. The hyperparameter settings are reported in Table 1. The total number of sampling steps is fixed to be 28 across all methods, which is the default value recommended by Flux.1[dev](Black Forest Labs, 2024).

## D. Extra Experimental Results

We provide additional results that complement the main paper, organized into qualitative comparisons, prompt-sensitivity analyses, seed variability, and limitations. Code is available at https://github.com/Z-Jianxin/

syncSDE-release#.

### D.1. Additional qualitative Results

Additional qualitative results are provided in Figure 7 and 10, where we present more examples of edits produced by Sync-SDE and comparisons with competing approaches, including pixel-wise difference maps. These results further demonstrate that Sync-SDE achieves edits well-aligned with the target prompt while preserving the source image structure, consistently producing localized and faithful modifications across diverse scenarios. Furthermore, we provide qualitative results in Figure 8 and 9 to demonstrate the global style-transfer capacity of sync-SDE.

### D.2. Qualitative results under different editing strength

Figures 11, 12, and 13 present a comprehensive qualitative comparison of Sync-SDE against the baselines under all hyperparameter settings specified in Table 1. Across all figures, Sync-SDE consistently produces edits that most faithfully follow the target prompt while preserving the structural integrity and fine-grained details of the source image.

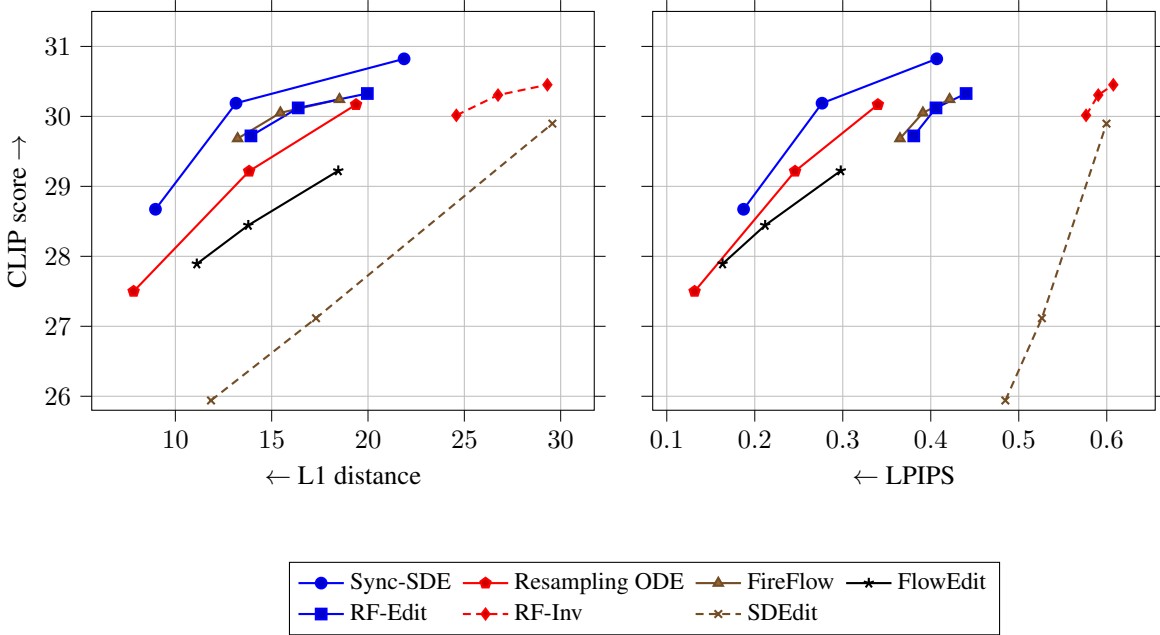

*Figure 6.* Trade-off between semantic alignment and perceptual similarity for different image editing methods. The x-axis reports distance metrics (L1 and LPIPS here), while the y-axis shows CLIP score. Points represent results for each method at different hyperparameter settings, and lines connect results from lower to higher distance. A higher CLIP score indicates better semantic consistency with the target prompt, while a lower distance means higher visual fidelity to the source image. Methods toward the upper-left corner achieve a better balance between preserving image structure and matching the edit prompt.

## D.3. Prompt Effects on Editing Performance

Figure 14 and Figure 15 qualitatively examine the role of prompt specificity and accuracy in editing performance for the tasks of adding glasses and replacing a spoon with a fork, respectively. The source prompts $c_{src,1-4}$ decrease in descriptive detail as the index increases, while $c_{src,5}$ and $c_{src,6}$ are intentionally misspecified to test the effect of source prompt accuracy. Similarly, the target prompts $c_{tar,1-4}$ form a hierarchy from very detailed to minimal. We list them here for completeness.

**Source prompts of Figure 14:**

- $c_{src,1}$ = "Portrait of a young woman with short dark hair, gazing directly at the camera, wearing a sheer black lace top with floral patterns. She leans slightly forward beside a reflective glass wall, soft natural light illuminating her face, blurred outdoor background with golden tones, cinematic shallow depth of field, fine detail."

- $c_{src,2}$ = "Close-up portrait of woman in black lace top, short dark hair, leaning by glass, looking at camera, warm sunlight background, shallow focus."

- $c_{src,3}$ = "Portrait of woman with short dark hair in lace clothing, leaning by window, soft background blur."

- $c_{src,4}$ = "Woman in lace top looking at camera."

- $c_{src,5}$ = "Portrait of a woman in a bright red dress with sequins, standing outdoors in front of a city skyline at night."

- $c_{src,6}$ = "Casual photo of woman in sportswear jogging on a beach at sunrise, waves in background."

**Target prompts of Figure 14:**

- $c_{tar,1}$ = "Portrait of a young woman with a pair of glasses and short dark hair, gazing directly at the camera, wearing a sheer black lace top with floral patterns. She leans slightly forward beside a reflective glass wall, soft natural light illuminating her face, blurred outdoor background with golden tones, cinematic shallow depth of field, fine detail."

- $c_{tar,2}$ = "Close-up portrait of woman with a pair of glasses in black lace top, short dark hair, leaning by glass, looking at camera, warm sunlight background, shallow focus."

- $c_{tar,3}$ = "Portrait of woman with a pair of glasses and short dark hair in lace clothing, leaning by window, soft background blur."

- $c_{tar,4}$ = "Woman with a pair of glasses in lace top looking at camera."

| Method | Hyperparameter Settings |
|---|---|
| Sync-SDE | source guidance = 1.0, target guidance = 1.0, starting index=4
source guidance = 1.5, target guidance = 1.5, starting index=4
source guidance = 2.5, target guidance = 2.5, starting index=4 |
| Resampling ODE | source guidance = 1.5, target guidance = 1.5, starting index=4
source guidance = 2.5, target guidance = 2.5, starting index=4
source guidance = 3.5, target guidance = 3.5, starting index=4 |
| FireFlow | guidance = 2, number of inject steps = 2, editing technique = replace_v
guidance = 2, number of inject steps = 3, editing technique = replace_v
guidance = 2, number of inject steps = 4, editing technique = replace_v |
| FlowEdit | source guidance = 1.5, target guidance = 3.5, $n_{min} = 0$, $n_{max} = 24$, $n_{avg} = 1$
source guidance = 1.5, target guidance = 4.5, $n_{min} = 0$, $n_{max} = 24$, $n_{avg} = 1$
source guidance = 1.5, target guidance = 5.5, $n_{min} = 0$, $n_{max} = 24$, $n_{avg} = 1$ |
| RF-Edit | Guidance = 2, number of inject steps = 2
Guidance = 2, number of inject steps = 3
Guidance = 2, number of inject steps = 4 |
| RF-Inv | target guidance = 3.5, stop index = 6, $\gamma = 0.5$, $\eta = 0.9$
target guidance = 3.5, stop index = 7, $\gamma = 0.5$, $\eta = 0.9$
target guidance = 3.5, stop index = 8, $\gamma = 0.5$, $\eta = 0.9$ |
| SDEdit | target guidance = 5.5, starting index = 7
target guidance = 5.5, starting index = 14
target guidance = 5.5, starting index = 21 |

*Table 1.* Hyperparameter configurations evaluated for each method. For each method, three representative settings are selected to probe the trade-off between semantic alignment and fidelity.

**Source prompts of Figure 15:**

- $c_{src,1}$ = "Minimalist coffee scene with small glass of dark espresso topped with golden crema, placed on rectangular wooden board. A silver spoon rests beside the glass. Background shows a clear glass holding napkins and cutlery, set against a light gray wall, tabletop in dark smooth finish, clean modern aesthetic, natural lighting."

- $c_{src,2}$ = "Glass of espresso with crema on wooden board, silver spoon beside, glass with napkins in background, minimalist modern café style."

- $c_{src,3}$ = "Small espresso glass on wooden board with spoon, simple background."

- $c_{src,4}$ = "Espresso in glass with spoon."

- $c_{src,5}$ = "Large ceramic teapot with green tea and a plate of cookies on wooden tray, cozy rustic kitchen scene."

- $c_{src,6}$ = "Outdoor picnic table with paper cup of cappuccino, croissant, and checkered cloth, bright sunny park."

**Target prompts of Figure 15:**

- $c_{tar,1}$ = "Minimalist coffee scene with small glass of dark espresso topped with golden crema, placed on rectangular wooden board. A silver fork rests beside the glass. Background shows a clear glass holding napkins and cutlery, set against a light gray wall, tabletop in dark smooth finish, clean modern aesthetic, natural lighting."

- $c_{tar,2}$ = "Glass of espresso with crema on wooden board, silver fork beside, glass with napkins in background, minimalist modern café style."

- $c_{tar,3}$ = "Small espresso glass on wooden board with fork, simple background."

- $c_{tar,4}$ = "Espresso in glass with fork."

The results show that when both source and target prompts are detailed and of comparable granularity, the edits are most faithful, preserving subject identity and contextual features. In contrast, misspecified or minimal prompts often lead to altered identities in Figure 14, and to lost textures of the wooden board, altered fine details on the napkins, and degraded coffee foam in Figure 15. In each figure, all images are generated with the same forward Brownian path.

**D.4. Effects of $t_0$**

Figure 16 qualitatively examines the role of $t_0$ in editing performance. By varying the starting time of the rectified flow, we observe a clear trade-off: smaller values of $t_0$ yield stronger, more comprehensive edits that more aggressively

follow the target prompt, while larger values of $t_0$ increasingly preserve the structure and appearance of the original image, resulting in more conservative edits.

### D.5. Effects of Discretization Size

To evaluate the robustness and efficiency of our approach, we analyze the effects of discretization size on the final editing quality. Figure 17 presents a visual comparison of edits performed using 7, 14, and 28 integration steps. Notably, SyncSDE holds strong performance across varying discretization steps. Even under highly accelerated sampling regimes with as few as 7 steps, the method successfully executes meaningful edits without compromising the structural integrity of the original source image. Increasing the step count to 14 or 28 maintains this high visual fidelity without introducing artifacts. This consistency demonstrates the stability of the SyncSDE formulation, proving it to be highly reliable for efficient, high-quality image editing with minimal computational overhead.

### D.6. Variations with Different Brownian Motion Paths

Figure 18 demonstrates the variability of sync-SDE across repeated runs for the same source–target prompt pairs. While the results highlight the model's ability to generate diverse yet semantically consistent edits, they also reveal certain caveats of our approach. For example, in the first row, the second repetition introduces a random artifact not present in the other outputs. In the second row, the second-to-last edited image shows an unreasonably large glass of milk, and in the fourth edited image the proportions are also distorted. Finally, in the last row, the third edited image alters the person's appearance in a noticeable way. These examples illustrate that although sync-SDE maintains strong alignment with prompts across seeds, it may occasionally produce undesirable variations and may require multiple runs to get the desired fidelity.

### D.7. Limitations of sync-SDE

Sync-SDE is not designed as a general instruction-following model. Instead, due to its formulation as a greedy optimal transport procedure, it tends to exploit existing structures in the source image to satisfy the target prompt. While this property can yield faithful and localized edits, it may also lead to suboptimal behavior depending on the use case. As illustrated in Figure 19, the dessert is altered to a different type rather than simply removing the specified fruits, the grass is covered with only a shallow layer of snow rather than a deep snow cover, and the potato is enlarged to fill the space where the salt was supposed to be removed. These examples highlight that sync-SDE preserves too much of the original structure when the task requires more radical changes. Similar issues also occur in other methods, such as FlowEdit (Kulikov et al., 2024), though our method generally produces more faithful edits even if it is not yet ideal.

### D.8. Additional Quantitative Comparisons

We perform quantitative evaluation on the Div2K dataset proposed in (Kulikov et al., 2024) and the PIE-Bench dataset (Ju et al., 2024). The results are shown in Figures 20 and 21, respectively. CSDE achieves the strongest trade-off on both benchmarks. Methods nearer the upper-left offer the best balance between preserving image structure and matching the edit prompt.

Recent pretrained instruction-based editing models such as Flux Kontext (Labs et al., 2025) and Qwen Image Edit (Wu et al., 2025) accept an input image alongside a natural language instruction and directly produce the edited output, bypassing explicit inversion. To evaluate these models on our benchmark of 306 source-target prompt pairs, we generate editing instructions from each (source prompt, target prompt) pair using Claude Sonnet 4.6 (Anthropic, 2025), which we found to produce the most reliable and semantically faithful instructions among the large language models we tested; we then manually polished the outputs to correct occasional ambiguities. We adopt guidance scales of 3.5 for Kontext and 4.0 for Qwen Edit, and sweep over $\{2.5, 3.5, 5.0\}$ and $\{3.0, 4.0, 5.0\}$ respectively to trace the trade-off between prompt adherence and source fidelity. As shown in Figure 22, Sync-SDE achieves a more favorable trade-off than both Kontext and Qwen Edit. Moreover, combining Sync-SDE with these instruction-based models further improves source similarity while maintaining competitive CLIP scores.

Figure 23 provides a qualitative comparison. Kontext and Qwen-Edit frequently alter regions unrelated to the editing instructions. When Sync-SDE is combined with either instruction-based model, the off-target changes are substantially reduced.

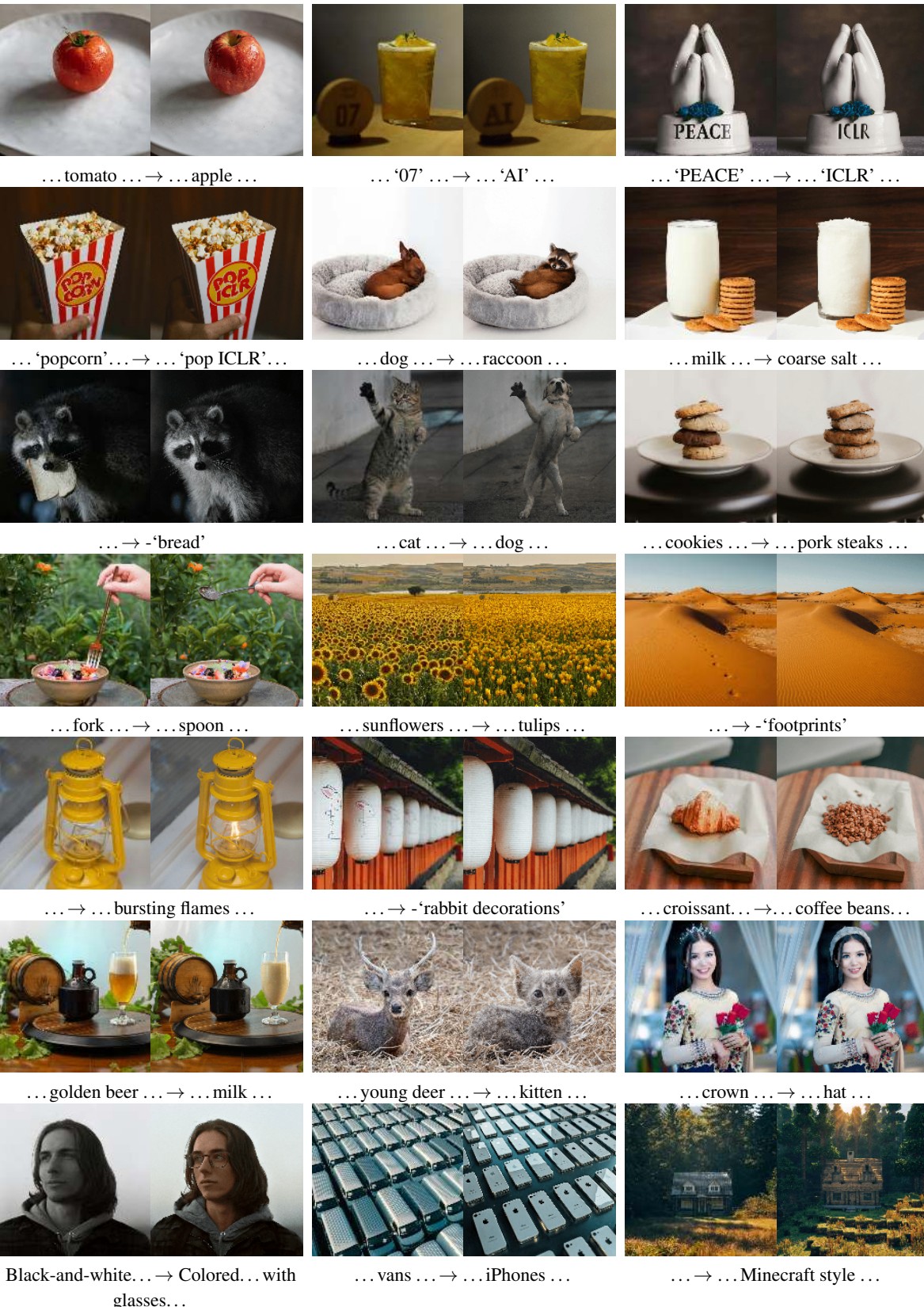

*Figure 7.* Each pair shows the source image on the left and the edited result on the right. The text below each pair specifies the shift from the source prompt to the target prompt. A leading minus sign ('-') indicates the use of a negative prompt.

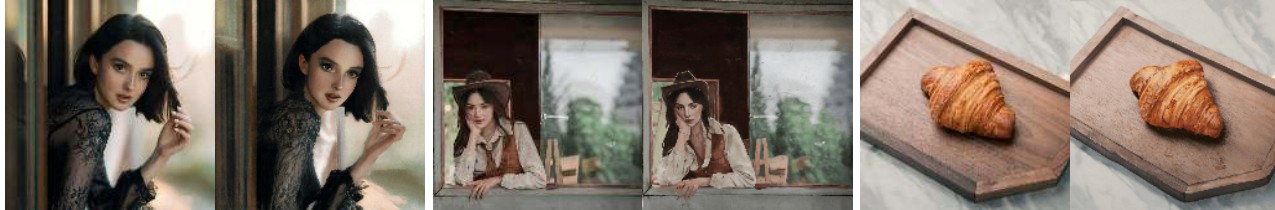

*Figure 8.* Each pair shows the source image on the left and the edited result on the right. Images are transformed from realistic images to oil-painting style.

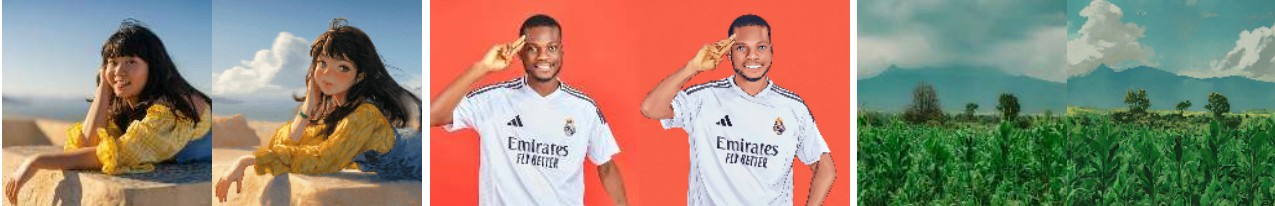

*Figure 9.* Each pair shows the source image on the left and the edited result on the right. Images are transformed from realistic images to anime style.

| Original | Sync-SDE | Fireflow | Flowedit | RF-Edit | RF-Inv | SDEdit |
|----------|----------|----------|----------|---------|--------|--------|

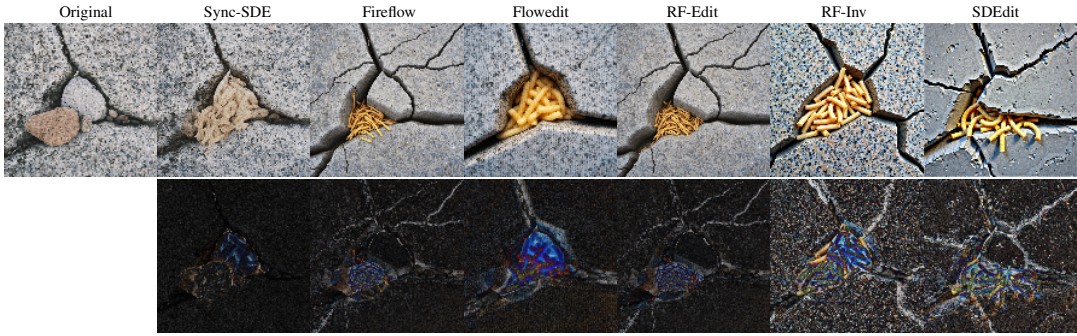

$c_{\mathrm{src}}$ =*A close-up of weathered stone with visible cracks, where small rocks and pebbles, including one tan and one gray, are nestled tightly within the crevices.*

$c_{\mathrm{tar}}$ =*A close-up of weathered stone with visible cracks, where many French fries are nestled tightly within the crevices.*

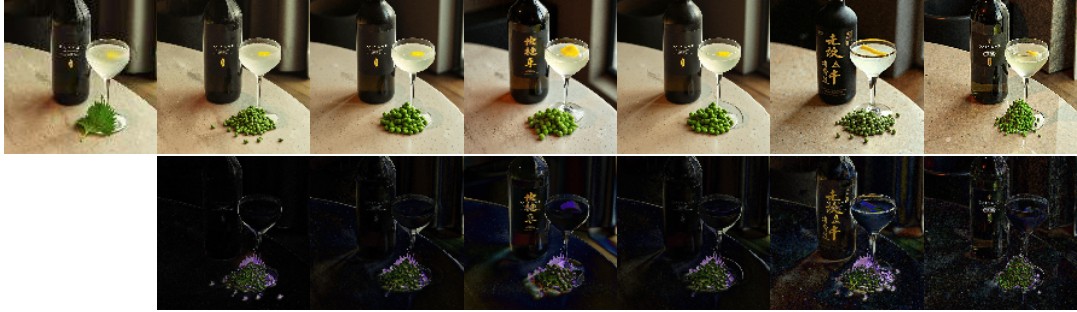

$c_{\mathrm{src}}$ =*Elegant bottle of Daiyame Japanese shochu beside a chilled cocktail glass with lemon twist, placed on a textured stone table with a fresh green shiso leaf.*

$c_{\mathrm{tar}}$ =*Elegant bottle of Daiyame Japanese shochu beside a chilled cocktail glass with lemon twist, placed on a textured stone table with a pile of green peas.*

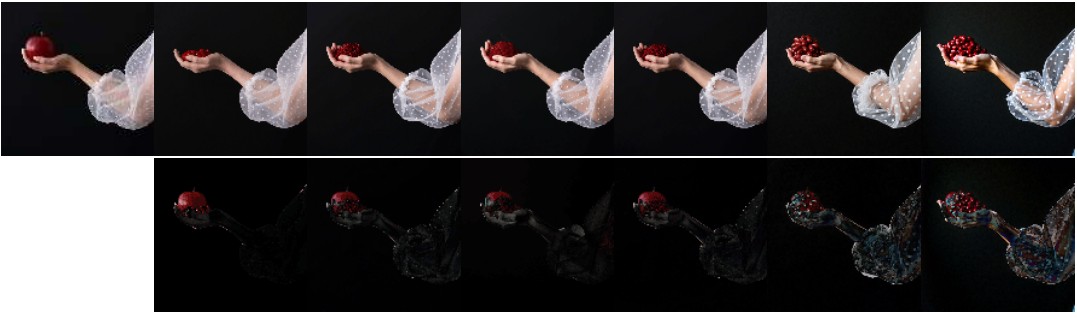

$c_{\mathrm{src}}$ =*A delicate hand in a sheer white polka-dotted sleeve holding a shiny red apple, posed against a solid black background.*

$c_{\mathrm{tar}}$ =*A delicate hand in a sheer white polka-dotted sleeve holding a pile of red beans, posed against a solid black background.*

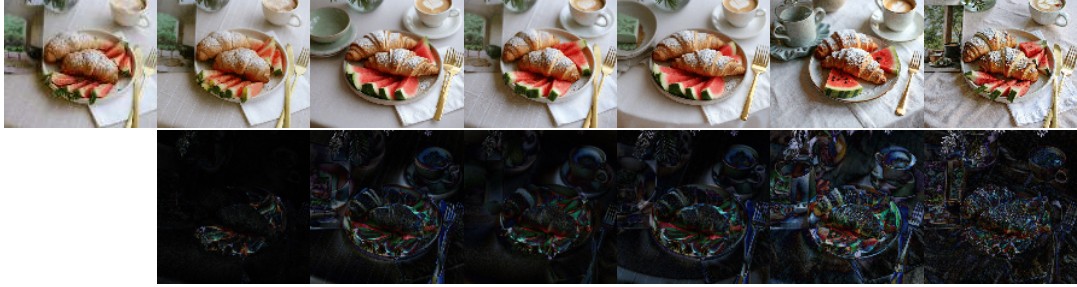

$c_{\mathrm{src}}$ =*A cozy breakfast scene with two croissants dusted with powdered sugar, served on a plate with fresh sliced strawberries, accompanied by a cup of cappuccino and golden cutlery on a light tablecloth.*

$c_{\mathrm{tar}}$ =*A cozy breakfast scene with two croissants dusted with powdered sugar, served on a plate with fresh sliced watermelons, accompanied by a cup of cappuccino and golden cutlery on a light tablecloth.*

*Figure 10.* Qualitative comparison of Sync-SDE with FireFlow (Deng et al., 2024), FlowEdit (Kulikov et al., 2024), RF-Edit (Wang et al., 2025a), RF-Inv (Rout et al., 2025a), and SDEdit (Meng et al., 2022). For each image, we show the original image followed by the edited results from each method. The next row shows the corresponding pixel-wise difference maps. Brighter regions indicate larger changes.

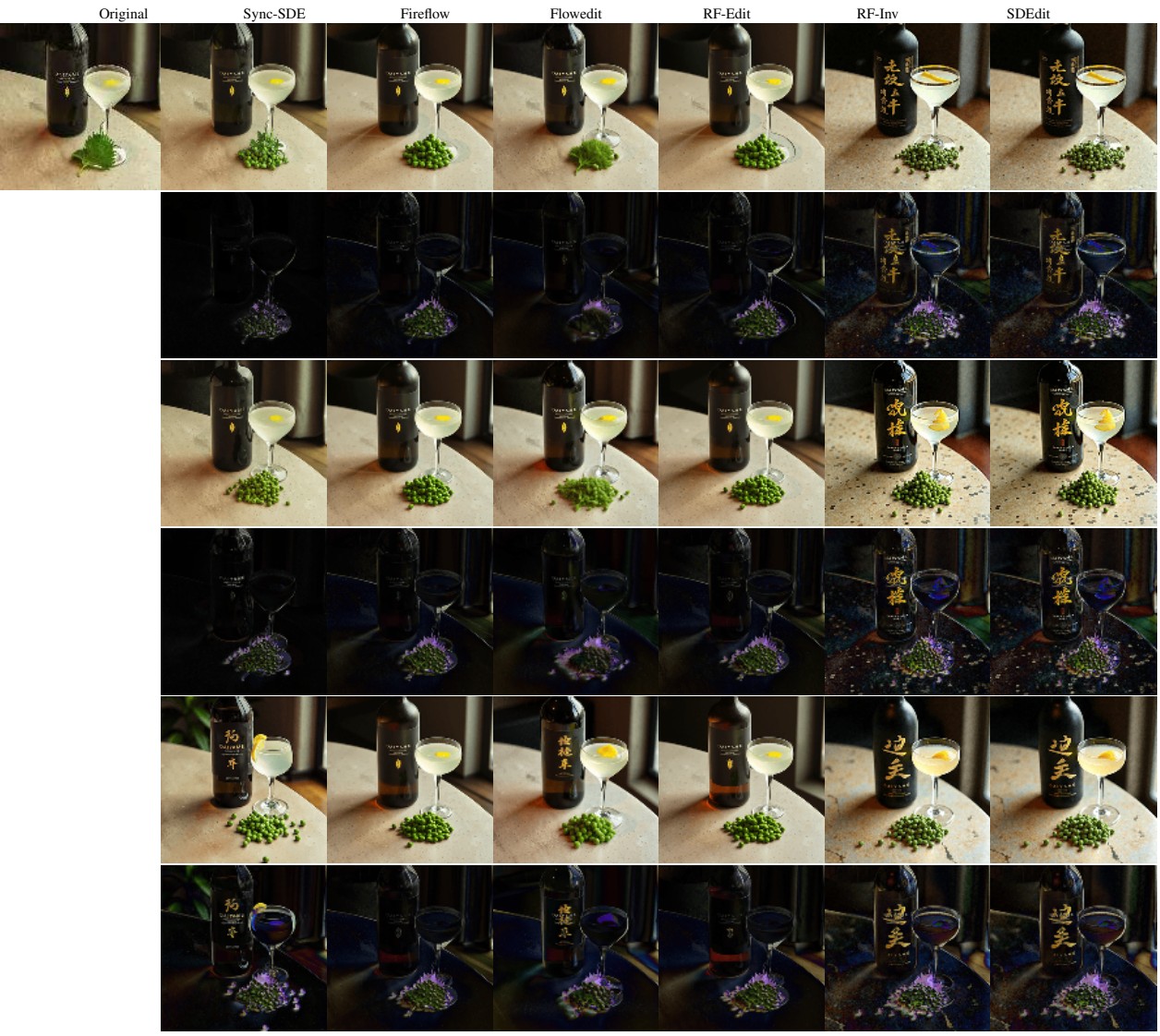

*Figure 11.* Qualitative comparison of Sync-SDE with recent semantic editing baselines: FireFlow (Deng et al., 2024), FlowEdit (Kulikov et al., 2024), RF-Edit (Wang et al., 2025a), RF-Inv (Rout et al., 2025a), and SDEdit (Meng et al., 2022) with all hyperparameter choices in Table 1. The editing strength increases from top to bottom. For each strength level, we show the outputs of all methods, with pixel-wise difference maps displayed directly below. Brighter regions in the difference maps indicate larger deviations from the source image, illustrating how each method trades off semantic change and structural preservation as the editing strength increases.

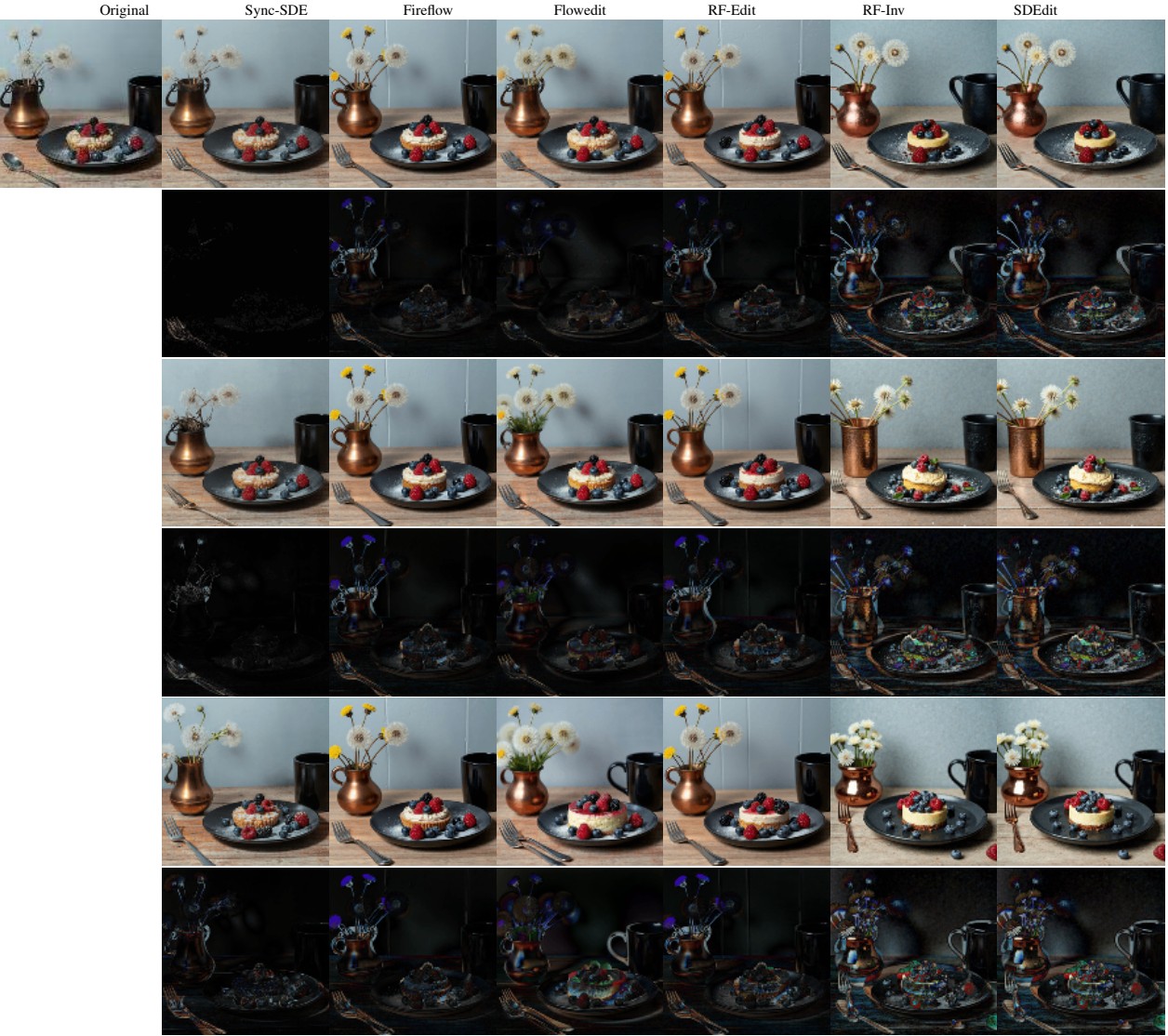

*Figure 12.* Qualitative comparison of Sync-SDE with recent semantic editing baselines: FireFlow (Deng et al., 2024), FlowEdit (Kulikov et al., 2024), RF-Edit (Wang et al., 2025a), RF-Inv (Rout et al., 2025a), and SDEdit (Meng et al., 2022) with all hyperparameter choices in Table 1. The editing strength increases from top to bottom. For each strength level, we show the outputs of all methods, with pixel-wise difference maps displayed directly below. Brighter regions in the difference maps indicate larger deviations from the source image, illustrating how each method trades off semantic change and structural preservation as the editing strength increases.

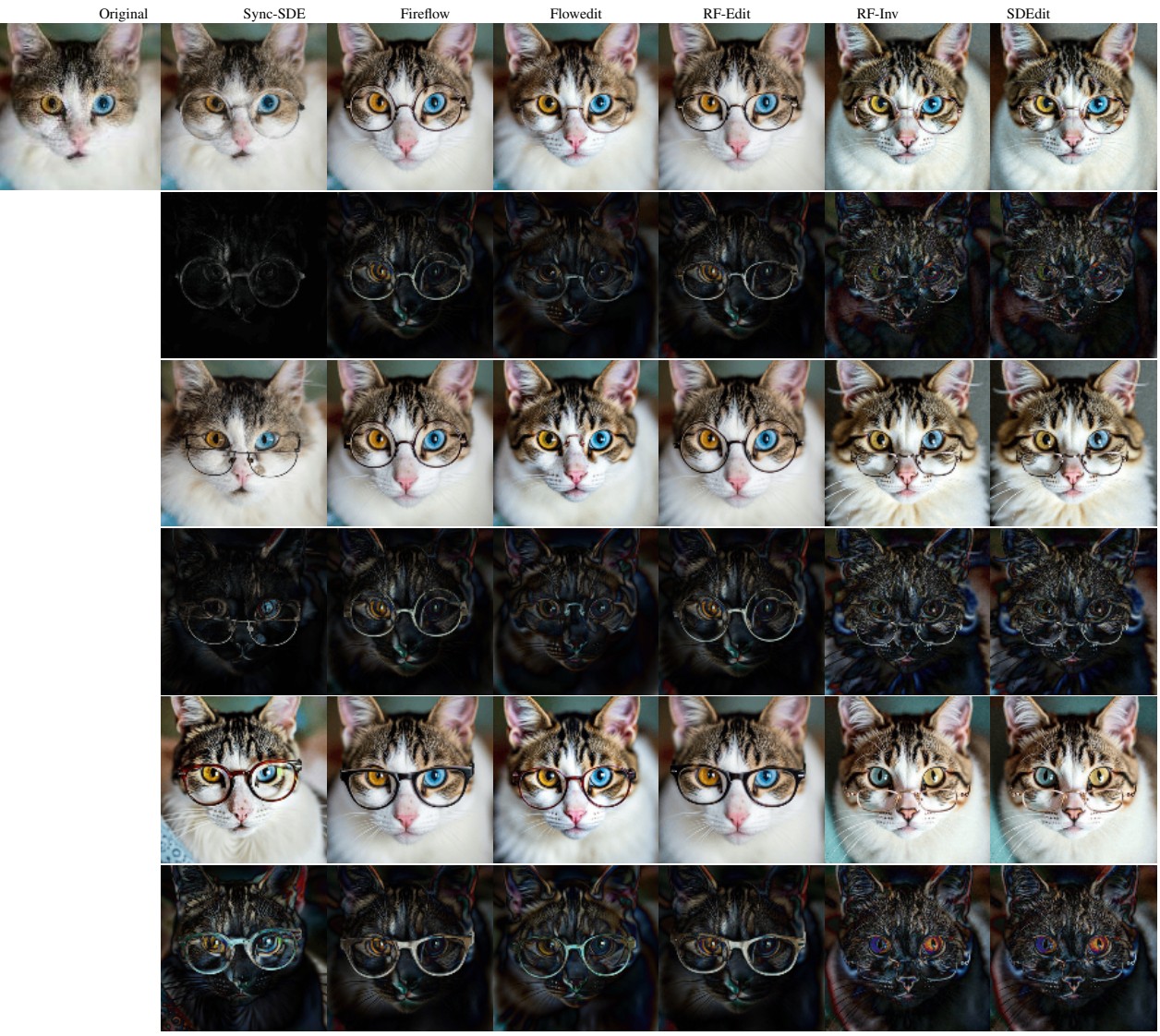

*Figure 13.* Qualitative comparison of Sync-SDE with recent semantic editing baselines: FireFlow (Deng et al., 2024), FlowEdit (Kulikov et al., 2024), RF-Edit (Wang et al., 2025a), RF-Inv (Rout et al., 2025a), and SDEdit (Meng et al., 2022) with all hyperparameter choices in Table 1. The editing strength increases from top to bottom. For each strength level, we show the outputs of all methods, with pixel-wise difference maps displayed directly below. Brighter regions in the difference maps indicate larger deviations from the source image, illustrating how each method trades off semantic change and structural preservation as the editing strength increases.

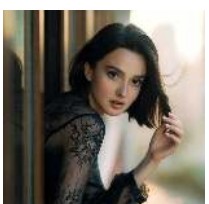

Original image

$c_{\mathrm{tar},1}$      $c_{\mathrm{tar},2}$      $c_{\mathrm{tar},3}$      $c_{\mathrm{tar},4}$

$c_{\mathrm{src},1}$

$c_{\mathrm{src},2}$

$c_{\mathrm{src},3}$

$c_{\mathrm{src},4}$

$c_{\mathrm{src},5}$

$c_{\mathrm{src},6}$

*Figure 14.* Editing study of sync-SDE on adding glasses to the subject in the original image (top). $c_{\mathrm{src},1-4}$ and $c_{\mathrm{tar},1-4}$ are progressively less detailed as the index increases from 1 to 4, while $c_{\mathrm{src},5}$ and $c_{\mathrm{src},6}$ are intentionally misspecified to test the impact of source prompt accuracy. Overall, edits obtained with both a detailed source prompt and a target prompt of comparable detail level yield the most successful results. All images are generated with the identical forward Brownian path. $c_{\mathrm{src},1-6}$ and $c_{\mathrm{tar},1-4}$ can be found in Section D.3

.

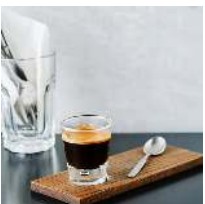

Original image

$c_{\text{tar},1}$      $c_{\text{tar},2}$      $c_{\text{tar},3}$      $c_{\text{tar},4}$

$c_{\text{src},1}$

$c_{\text{src},2}$

$c_{\text{src},3}$

$c_{\text{src},4}$

$c_{\text{src},5}$

$c_{\text{src},6}$

*Figure 15.* Editing study of sync-SDE on replacing a spoon with a fork in the original image (top). $c_{\text{src},1-4}$ and $c_{\text{tar},1-4}$ are progressively less detailed as the index increases from 1 to 4, while $c_{\text{src},5}$ and $c_{\text{src},6}$ are intentionally misspecified to test the impact of source prompt accuracy. Overall, edits obtained with both a detailed source prompt and a target prompt of comparable detail level yield the most successful results. All images are generated with the identical forward Brownian path. $c_{\text{src},1-6}$ and $c_{\text{tar},1-4}$ can be found in Section D.3
.

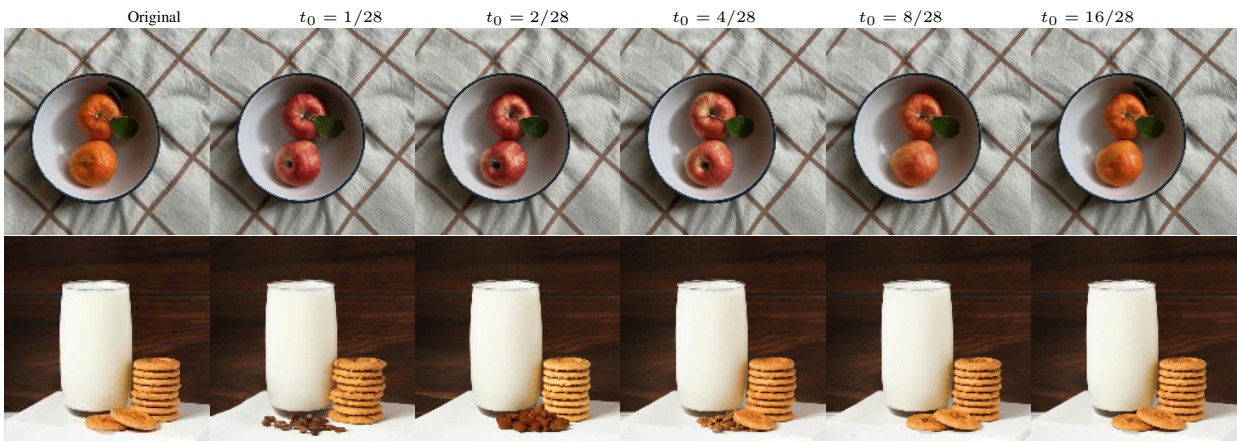

*Figure 16.* Qualitative effect of varying $t_0$ on editing behavior. **Top row:** The intended prompt is to replace the oranges with apples. **Bottom row:** The intended prompt is to replace the two cookies in front with coffee beans.

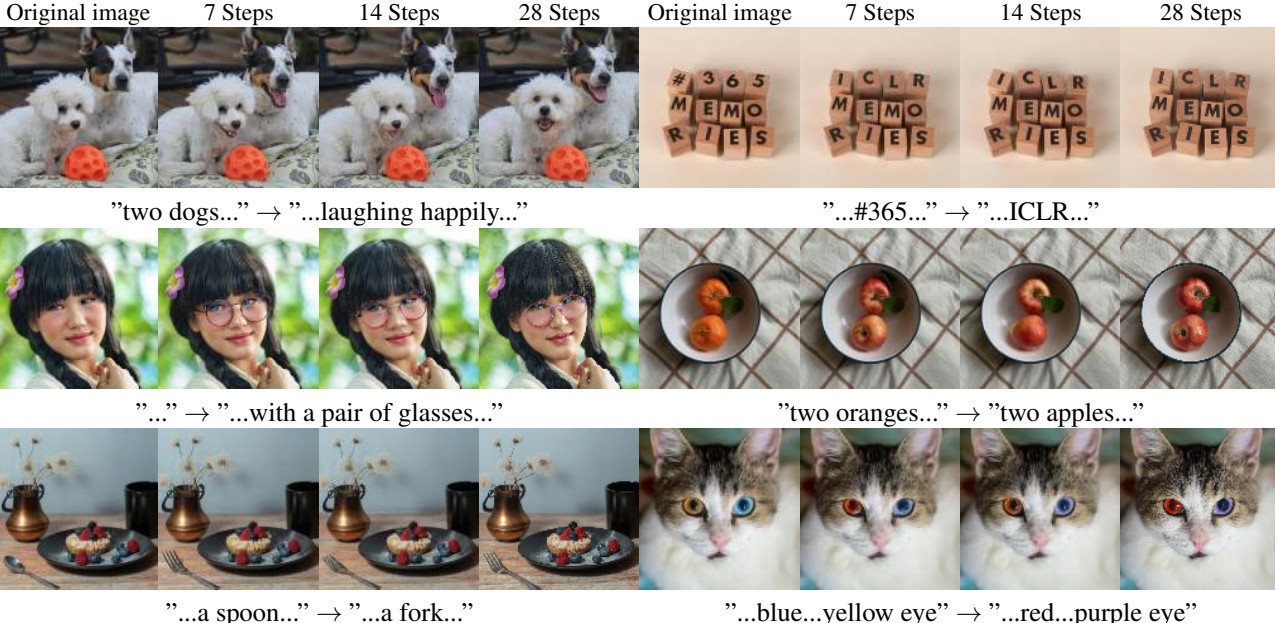

*Figure 17.* Qualitative results demonstrating the effects of discretization size on image editing. We compare the outputs of syncSDE using 7, 14, and 28 steps alongside the original source images and their respective prompt modifications. The results illustrate that syncSDE holds strong performance across all discretization steps, successfully applying semantic changes and preserving the original structure even at highly accelerated regimes (7 steps). All examples were generated with Flux.1[dev] (Black Forest Labs, 2024).

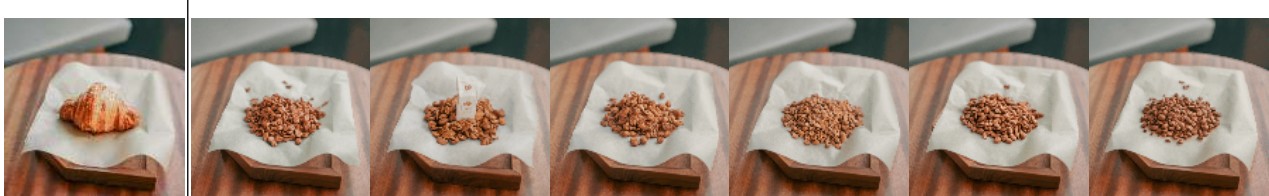

$c_{\mathrm{src}}$ =*Golden brown croissant with visible flaky layers resting on a sheet of white parchment paper. The pastry sits on a wooden tray placed on a round wooden table, softly lit by natural daylight. Background is softly blurred.*

$c_{\mathrm{tar}}$ =*A pile of brown whole coffee beans resting on a sheet of white parchment paper. The coffee beans sit on a wooden tray placed on a round wooden table, softly lit by natural daylight. Background is softly blurred.*

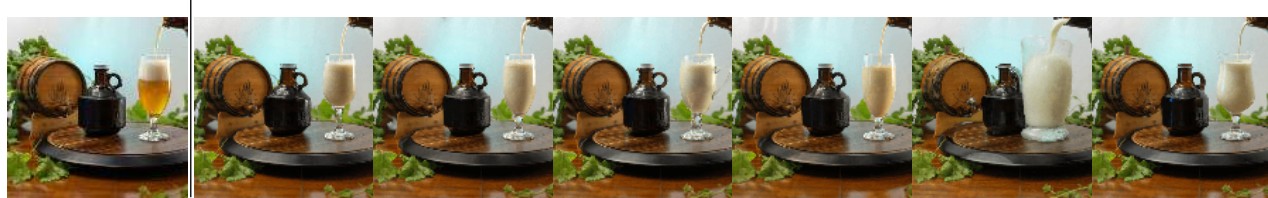

$c_{\mathrm{src}}$ =*Glass of golden beer being poured, topped with frothy foam, placed on a wooden tray. Beside it is a dark brown glass jug with handle, and in the background a small wooden beer barrel with leaf vines draped around it. Scene is softly lit with a clean backdrop, high detail.*

$c_{\mathrm{tar}}$ =*Glass of milk being poured, placed on a wooden tray. Beside it is a dark brown glass jug with handle, and in the background a small wooden beer barrel with leaf vines draped around it. Scene is softly lit with a clean backdrop, high detail.*

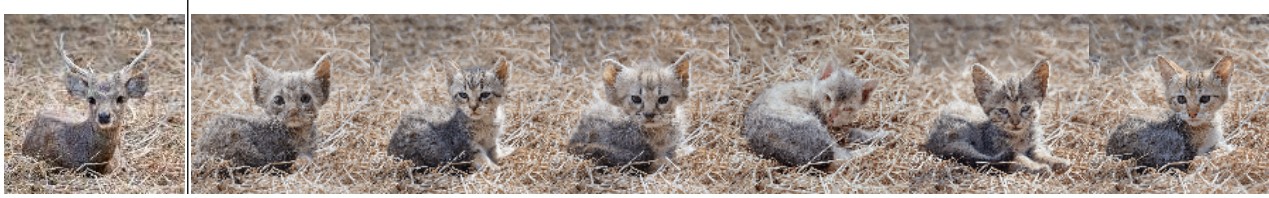

$c_{\mathrm{src}}$ = *Close-up of a young deer with short antlers resting on a bed of dry straw. The animal faces forward with calm, alert expression, ears perked and fur in warm brown tones. Sunlight highlights the texture of its coat and the straw around it. Natural wildlife portrait, rustic and serene atmosphere, high detail and photorealistic style.*

$c_{\mathrm{tar}}$ =*Close-up of a kitten resting on a bed of dry straw. The animal faces forward with calm, alert expression, ears perked and fur in warm brown tones. Sunlight highlights the texture of its coat and the straw around it. Natural wildlife portrait, rustic and serene atmosphere, high detail and photorealistic style.*

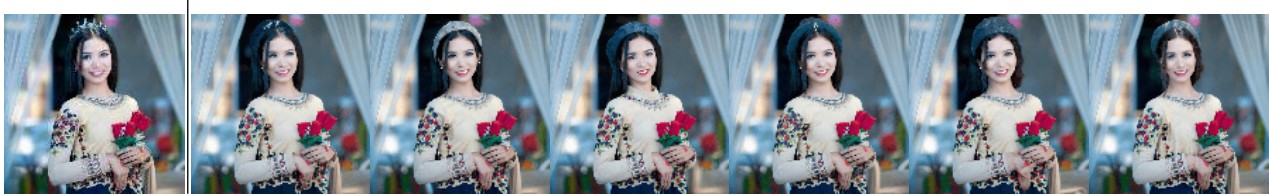

$c_{\mathrm{src}}$ =*Portrait of a young woman wearing a crown and traditional embroidered dress with floral patterns. She holds a bouquet of red roses in her hands and smiles warmly at the camera. The background is softly blurred with flowing white drapes framing the scene, creating a regal and festive atmosphere, high detail and vibrant colors.*

$c_{\mathrm{tar}}$ =*Portrait of a young woman wearing a hat and traditional embroidered dress with floral patterns. She holds a bouquet of red roses in her hands and smiles warmly at the camera. The background is softly blurred with flowing white drapes framing the scene, creating a regal and festive atmosphere, high detail and vibrant colors.*

*Figure 18.* Multiple independent runs of sync-SDE edits for four source-target prompt pairs. In each row, the leftmost image is the original image, followed by six edited results from different random seeds. The source and target prompts ($c_{\mathrm{src}}$ and $c_{\mathrm{tar}}$) are shown below each row. The examples demonstrate both the consistency and variability of sync-SDE across repeated generations.

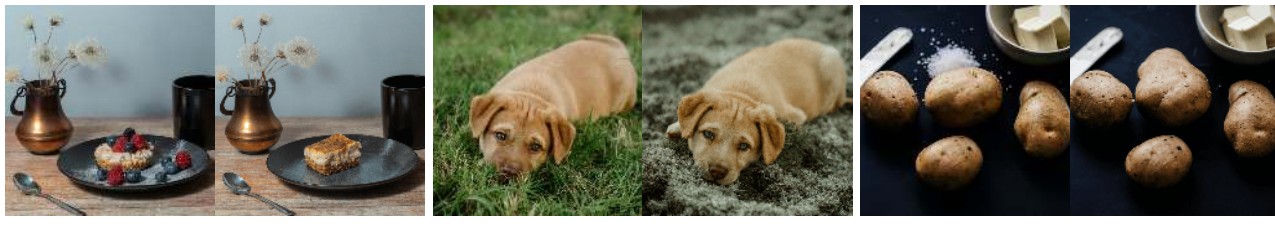

... → -'berries and blueberries'    ... green grass... → ... snow-covered land...    ... → -'a small pile of coarse salt'

*Figure 19.* Each pair shows the source image on the left and the edited result on the right. The text below each pair specifies the shift from the source prompt to the target prompt. A leading minus sign ('-') indicates the use of a negative prompt.

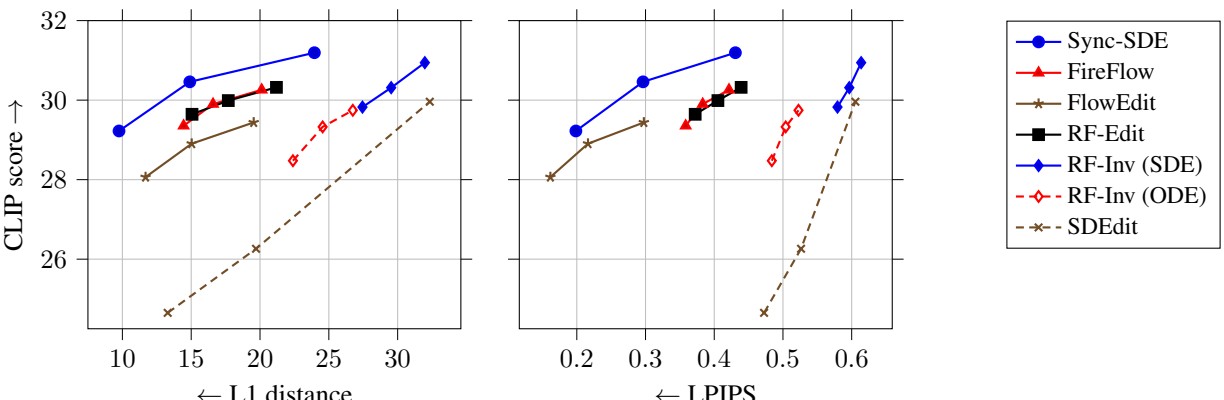

*Figure 20.* Trade-off between semantic alignment and perceptual similarity for different image editing methods on the Div2k dataset proposed in (Kulikov et al., 2024). The x-axis reports distance metrics (L1 and LPIPS here), while the y-axis shows CLIP score. Points represent results for each method at different hyperparameter settings, and lines connect results from lower to higher distance. A higher CLIP score indicates better semantic consistency with the target prompt, while a lower distance means higher visual fidelity to the source image. Methods toward the upper-left corner achieve a better balance between preserving image structure and matching the edit prompt.

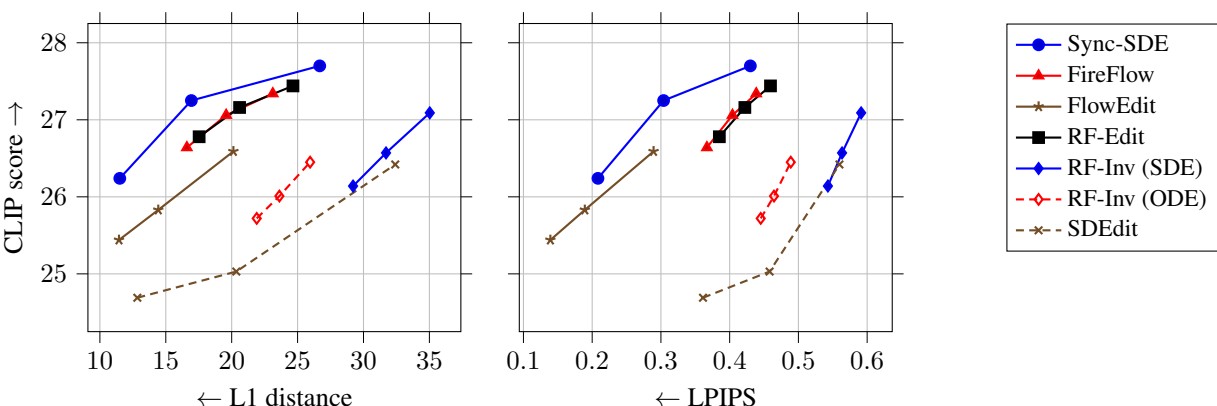

*Figure 21.* Trade-off between semantic alignment and perceptual similarity for different image editing methods on the PIE-Bench dataset (Ju et al., 2024). The x-axis reports distance metrics (L1 and LPIPS), while the y-axis shows CLIP score. Points represent results for each method at different hyperparameter settings, and lines connect results from lower to higher distance. A higher CLIP score indicates better semantic consistency with the target prompt, while a lower distance means higher visual fidelity to the source image. Methods toward the upper-left corner achieve a better balance between preserving image structure and matching the edit prompt.

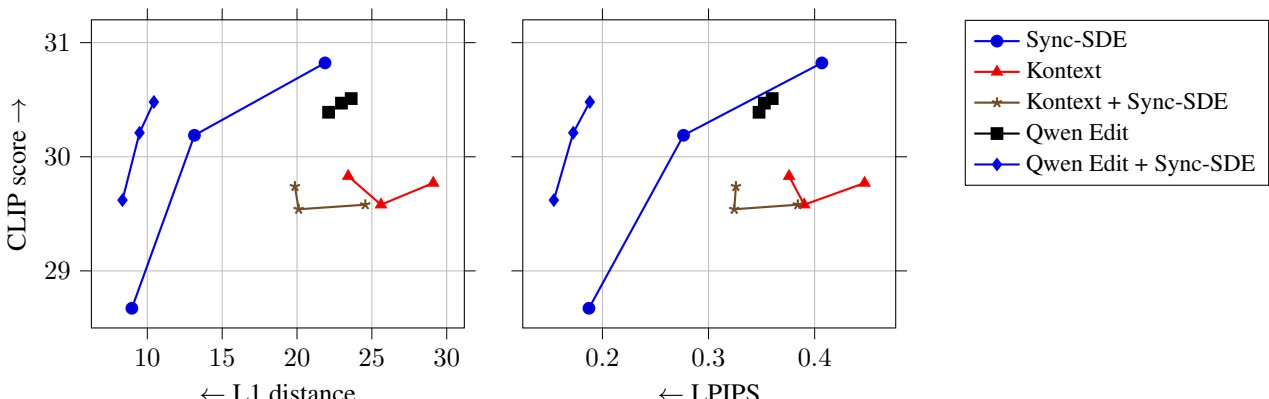

*Figure 22.* Trade-off between semantic alignment and perceptual similarity. The x-axis reports distance metrics (L1 and LPIPS), while the y-axis shows CLIP score. Points represent results at different hyperparameter settings, and lines connect results from lower to higher distance. Methods toward the upper-left corner achieve a better balance between preserving image structure and matching the edit prompt. Combining the Sync-SDE technique with Flux Kontext and Qwen Image Edit consistently improves the stability and similarity to the original image, shifting the operating curves toward lower perceptual distance while maintaining competitive CLIP scores.

Original  Sync-SDE  FluxKontext  FluxKontext+SyncSDE  QwenEdit  QwenEdit+SyncSDE

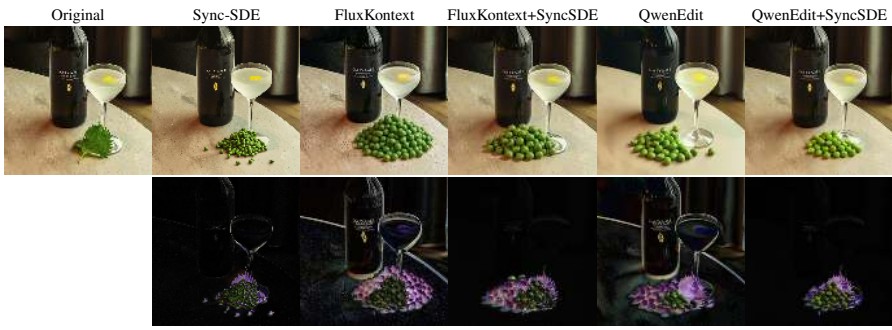

$c_{\mathrm{src}}$ =*Elegant bottle of Daiyame Japanese shochu beside a chilled cocktail glass with lemon twist, placed on a textured stone table with a fresh green shiso leaf.*

$c_{\mathrm{tar}}$ =*Elegant bottle of Daiyame Japanese shochu beside a chilled cocktail glass with lemon twist, placed on a textured stone table with a pile of green peas.*

$c_{\mathrm{instr}}$ =*Change fresh shiso leaf to pile of peas*

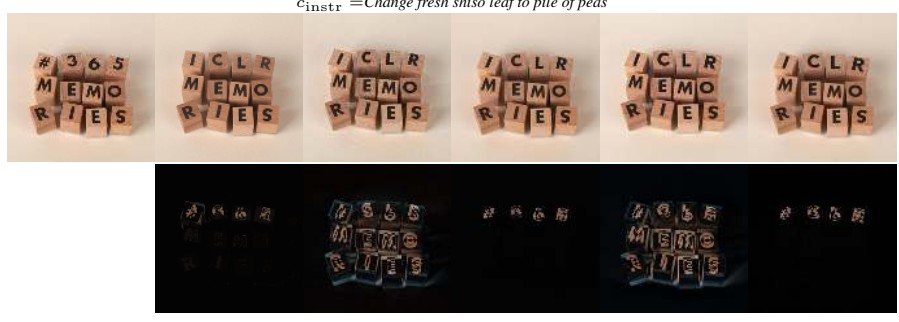

$c_{\mathrm{src}}$ =*there is a pile of wooden blocks that have the words '#365 Memories' written on them in black letters, sitting next to each other*

$c_{\mathrm{tar}}$ =*there is a pile of wooden blocks that have the words 'ICLR Memories' written on them in black letters, sitting next to each other*

$c_{\mathrm{instr}}$ =*Change #365 to ICLR*

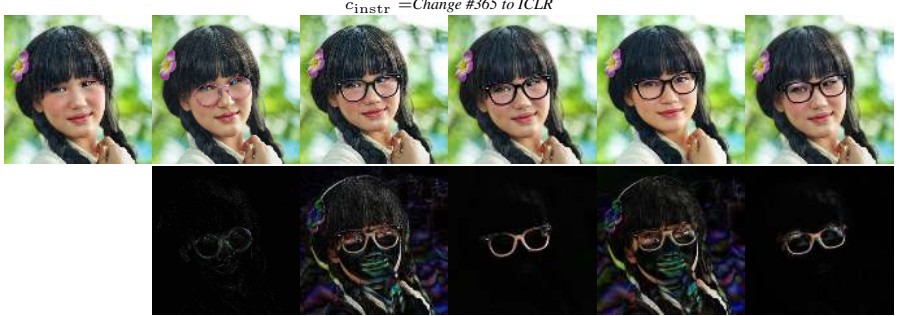

$c_{\mathrm{src}}$ =*A close-up portrait of a woman with long dark braided hair, wearing a white top, a purple and yellow plumeria flower tucked in her hair.*

$c_{\mathrm{tar}}$ =*A close-up portrait of a woman with a pair of glasses and long dark braided hair, wearing a white top, a purple and yellow plumeria flower tucked in her hair.*

$c_{\mathrm{instr}}$ =*Add a pair of glasses to the woman.*

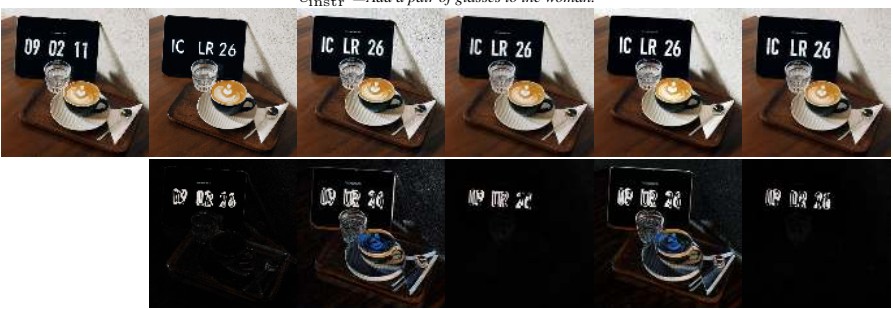

$c_{\mathrm{src}}$ =*A latte with latte art in a black cup on a saucer, served with a glass of water and a spoon on a wooden tray, next to a digital clock display reading "09 02 11".*

$c_{\mathrm{tar}}$ =*A latte with latte art in a black cup on a saucer, served with a glass of water and a spoon on a wooden tray, next to a digital clock display reading "IC LR 26".*

$c_{\mathrm{instr}}$ =*Change the digital clock display from '09 02 11' to 'IC LR 26'.*

*Figure 23.* Qualitative comparison of Sync-SDE with Flux Kontext (Labs et al., 2025), Kontext combined with Sync-SDE, Qwen Image Edit (Wu et al., 2025), and Qwen Edit combined with Sync-SDE. For each image, we show the original image followed by the edited results from each method. The next row shows the corresponding pixel-wise difference maps. Brighter regions indicate larger changes.

