# OpenReview forum: "Semantic Editing with Coupled Stochastic Differential Equations"
_ICML.cc/2026/Conference — ICML 2026 regular_

### Official Review · Reviewer_5Soj · 2026-03-03

**Soundness:** 3
**Presentation:** 3
**Significance:** 3
**Originality:** 3
**Overall Recommendation:** 4
**Confidence:** 4

**Summary:**

This paper introduces sync-SDE, a training-free, optimization-free approach to text-guided semantic image editing by coupling the reference and edited reverse-time generative processes. The central mechanism is to reuse the same backward stochastic noise path, constructed via time-reversal, for both trajectories, so edits arise primarily from the prompt-conditioned drift while stochasticity is aligned, yielding strong structural preservation with meaningful semantic change. The paper further motivates the design through a bicausal optimal transport view, arguing synchronous coupling is locally optimal under a quadratic deviation objective. Experiments show favorable preservation–edit trade-offs against several recent baselines.

**Compliance With Llm Reviewing Policy:**

Affirmed.

**Final Justification:**

The additional experiment results partially address my questions. But the detailed setting and human evaluation are still missing. Overall, I think this is a borderline paper, some amount of work is still needed to help it be a complete work. I hold my rating as weak accept, but strongly suggest the authors address these issues in the final version.

**Key Questions For Authors:**

1. Beyond Flux.1[dev] at 28 steps, how stable are results for different samplers / step counts?

2. How sensitive is sync-SDE to c_src mismatch, and can the method be made robust when c_src is noisy? The paper explicitly flags lack of guarantees otherwise.

3. Are baselines tuned under comparable guidance strength and compute budgets? Please report matched budgets.

4. Can the authors evaluate on established real-image editing benchmarks and add human studies for edit success vs faithfulness?

**Limitations:**

1.	Best suited for structure-preserving edits. Large geometry/layout changes likely require weakening coupling and may reduce quality.
2.	Performance may be sensitive to the generative model and discretization details despite the solver-agnostic narrative.
3.	Current evidence does not fully establish broad real-world robustness (diverse images, imperfect prompts, complex edits).

**Strengths And Weaknesses:**

Strengths

1.	Synchronous coupling via shared backward noise is simple, model-agnostic, and practically appealing.
2.	The method is straightforward to implement and does not rely on test-time optimization or extra networks.
3.	The bicausal OT interpretation is aligned with the method rather than being purely ornamental.
4.	Strong qualitative behavior on localized edits. Changes appear better confined to semantically relevant regions relative to baselines.

Weaknesses

1.	The dataset scale (306 triplets) and construction (curated prompts/images) limits confidence in generality.
2.	Reliance on CLIP-based adherence and perceptual metrics without human preference or standardized editing benchmarks leaves room for metric gaming.
3.	The transport view is explicitly stated to provide no guarantee if the source prompt does not describe the source image. This is a nontrivial practical limitation given real images often do not match text descriptions perfectly.
4.	The proposed methods run in comparable time (15–25s on H100), but does not provide a tighter accounting of score evaluations/overhead induced by coupling and backward-noise construction relative to baselines..
Key Questions

---

> ### Author Rebuttal · Authors · 2026-03-30
>
> We thank the reviewer for the thoughtful and constructive feedback, and for recognizing the simplicity and practical appeal of our synchronous coupling mechanism, the non-ornamental nature of the bicausal OT interpretation, and the strong qualitative behavior on localized edits. We address each concern below.
>
> ---
>
> ## Weakness 1: Dataset Scale
>
> We agree that a larger-scale evaluation would further strengthen the paper. That said, we note that the scale of our evaluation is consistent with (and in several cases exceeds) that of prior training-free editing methods in this domain. The table below summarizes evaluation dataset sizes from representative training-free editing works, all already cited in our paper:
>
>
> | Method | Dataset Size |
> | --- | --- |
> | Prompt-to-Prompt (Hertz et al., 2023) | < 100, Qualitative results only|
> | SEGA (Brack et al., 2023) | 250 |
> | FlowEdit (Kulikov et al., 2024) | 250 |
> | Fireflow (Deng et al., 2024) | 700 |
> | **Sync-SDE (Ours)** | 306 (our new constructed dataset) + 250 (div2k, in supplementary) = 556  |
>
> ---
>
> ## Weakness 2 & Question 4: Metrics and Standardized Benchmarks
>
> We agree that human evaluation is valuable. We'd like to note, however, that human evaluation is not standard for recent state-of-the-art training-free semantic editing papers. Among the best-performing training-free methods (FireFlow, FlowEdit, RF-Edit, RF-Inv, SDEdit), only RF-Edit provides human evaluation.
>
> Due to a recent institutional move, we are actively assessing our budget for a user study and will include one if financially feasible.
>
> ---
>
> ## Weakness 3 & Question 2: Sensitivity to Source Prompt Mismatch
>
> We appreciate the reviewer highlighting this important practical consideration. We would like to point out that **Figures 14 and 15 in the appendix directly address this concern**. In these figures, we explicitly test mis-specified source prompts by systematically varying the alignment between the source prompt and the actual source image. The qualitative results demonstrate that high-quality edits can still be produced even when the source prompt is inaccurate, as long as the source and target prompts are not extremely mis-aligned. We agree this is a meaningful limitation and will discuss it more explicitly in the revised manuscript. We are also happy to provide additional examples exploring prompt misalignment if the reviewer finds it helpful.
>
> ---
>
> ## Weakness 4: Computational Overhead of Coupling and Backward-Noise Construction
>
> We appreciate this question. The coupling and backward-noise construction steps involve only lightweight algebraic operations on the noise vectors and **do not require any additional neural network evaluations**. As a result, they introduce nearly zero computational overhead relative to standard sampling. All methods in our comparison use Flux with the same 28-step discretization, so the dominant cost (score function evaluations) is matched across the board. We will include explicit wall-clock time measurements in the revised paper to make this point precise.
>
> ---
>
> ## Question 1: Stability Across Different Samplers and Step Counts
>
> We agree this is an important question for establishing the generality of our approach. We will provide additional results with varying step counts and sampler configurations in the revised manuscript to address this concern.
>
> ---
>
> ## Question 3: Comparable Baselines and Compute Budgets
>
> Yes, all baselines are tuned under comparable conditions. We select hyperparameters for each baseline (including guidance strength and starting time) as recommended by their respective original papers, codebases, or GitHub releases. All methods use the same backbone (Flux) with the same 28-step discretization, ensuring matched computational budgets. These details are documented in Section C and Table 1 in the appendix. We will add a more prominent discussion of this matched-budget setup in the main paper to make it immediately clear to readers.

---

> > ### Author Rebuttal · Reviewer_5Soj · 2026-04-04
> >
> > I thank the authors for their rebuttal. However, only a limited portion of my concerns has been adequately addressed. For Weakness 1, the clarification on dataset scale is helpful, but I still do not see a direct response regarding real-image editing. For the human evaluation, citing prior works as common practice is not, by itself, sufficient to address my concern.
> >
> > For Figures 14 and 15, the source and target prompts are not specified in enough detail, making the effectiveness of these results difficult to judge. In addition, for Weakness 4 and Question 1, the rebuttal still lacks quantitative results on time cost. Overall, given the incomplete and somewhat coarse rebuttal, I am inclined to downgrade my current score to weak reject.

---

> > > ### Author Response · Authors · 2026-04-07
> > >
> > > Thank you for your constructive feedback and for your patience while we ran these additional experiments. All of the following changes are already integrated in the paper:
> > >
> > > # Scale and Real-image Editing:
> > >
> > > We present the quantitative experimental results on PIE-Bench [3]:
> > >
> > > [PIE-Bench Results](https://anonymous.4open.science/r/random-3580/pie_plot.png)
> > >
> > > The setup is the same as the experiments in the main paper: we run experiments on three hyperparameter settings as recommended in their papers or codebase, and we plot the L1 distance vs. CLIP score and LPIPS vs. CLIP score trade-offs. From the plot, we can see that Sync-SDE continues to achieve a highly favorable trade-off on PIE-Bench.
> > >
> > >
> > >
> > >
> > > # Effects of Prompt Quality:
> > >
> > > Regarding Figures 14 and 15, the **exact prompts** are **already** detailed in **Section D.3** of the appendix (**lines 749-769 on page 14**, and **lines 795-824 on page 15**). To make this clearer for readers, we have added pointers to the figure captions.
> > >
> > >
> > >
> > >
> > > # Run time
> > >
> > > We ran Sync-SDE and all baseline methods on 100 1024x1024 images and report their average run times on a single H100 below:
> > >
> > > | Method | Time |
> > > | :--- | :--- |
> > > | Sync-SDE | 15.6s |
> > > | FireFlow | 15.3s |
> > > | FlowEdit | 15.0s |
> > > | RFEdit | 24.9s |
> > > | RFInv (ODE) | 15.6s |
> > > | RFInv (SDE) | 16.2s |
> > > | SDEdit | 14.7s |
> > >
> > >
> > >
> > >
> > > # Different Step Counts
> > >
> > > The figure below presents a visual comparison of edits performed using 7, 14, and 28 integration steps:
> > >
> > > [Visual Comparison of Integration Steps](https://anonymous.4open.science/r/random-3580/steps.png)
> > >
> > > Notably, Sync-SDE maintains strong performance across varying discretization steps. Even under highly accelerated sampling regimes with as few as 7 steps, the method successfully executes meaningful edits without compromising the structural integrity of the original source image.
> > >
> > >
> > >
> > >
> > > # Human evaluation:
> > >
> > > We fully agree that human evaluation is an important metric. We have designed the survey and prepared the data:
> > >
> > > [Survey Preview](https://anonymous.4open.science/r/random-3580/survey.png)
> > >
> > > However, our MTurk account recently encountered a credit limit issue, which appears to be a common restriction for new users. We will complete the human evaluation and share the results as soon as our account is unblocked.

---

### Official Review · Reviewer_fdn3 · 2026-03-12

**Soundness:** 3
**Presentation:** 4
**Significance:** 2
**Originality:** 2
**Overall Recommendation:** 4
**Confidence:** 4

**Summary:**

This paper proposes Sync-SDE, a training-free semantic image editing method leveraging coupled stochastic differential equations (SDEs) and shared correlated noise to guide sampling of pretrained generative models. Grounded in Anderson's time-reversal theorem and optimal transport theory, it is evaluated on a self-constructed dataset.

**Compliance With Llm Reviewing Policy:**

Affirmed.

**Final Justification:**

The authors have addressed my two primary concerns during the discussion period: (1) evaluation on PIE-Bench now demonstrates favorable trade-offs on an established benchmark, and (2) quantitative and qualitative comparisons against FluxKontext and Qwen-Edit substantiate the method's competitiveness, even against trained models. The additional runtime and step-count ablation results are also appreciated.

I still consider the core contribution—coupling via shared reverse Brownian motion—to be a relatively natural extension of existing noise-reuse ideas, and the pending human evaluation is a minor gap. However, the revised empirical evidence is now sufficient to support the paper's claims. I raise my score to weak accept.

**Key Questions For Authors:**

1. Please compare Sync-SDE directly against Qwen-Image-Edit [1], Kontext [2], and other relevant SDE-based editing methods, reporting consistent metrics (CLIP, LPIPS, L1) and runtime. Addressing this is essential for the significance concern.

2. Please evaluate on dedicated semantic editing benchmarks—PIE-Bench [3], ImgEdit [4], and GEdit-Bench [5]—covering complex real-world editing tasks (large-scale structural changes, multi-semantic editing), and present all hyperparameter settings in the main paper to eliminate cherry-picking perceptions.

3. Can Sync-SDE's core mechanisms be applied as a plug-and-play module on top of trained editing models (Qwen-Image-Edit [1], Kontext [2]) to boost fidelity preservation? Addressing this would substantially broaden the method's impact.

Addressing points 1 and 2 is essential to substantiate practical value.

**Limitations:**

1. No comparison with mainstream open-source trained SOTA (Qwen-Image-Edit [1], Kontext [2]).

2. Narrow and inappropriate evaluation: the public benchmark used is not designed for semantic editing; no testing on dedicated benchmarks (PIE-Bench [3], ImgEdit [4], GEdit-Bench [5]).

3. Selective result presentation (best-of-3 hyperparameter results in main paper) creates cherry-picking perceptions. (see Line 310-312, "For each example and for each method compared, we select, among the three hyperparameter settings reported in the quantitative result")

4. No fundamentally new technical contributions beyond standard adaptation of existing coupled SDE and noise reuse ideas.

**Strengths And Weaknesses:**

### Strengths

- The mathematical formulation and algorithm design are technically sound within the training-free SDE framework, and the theoretical arguments appear well grounded.

- The paper is clearly written and well-structured, with a coherent narrative and a logical progression from motivation to formulation and experiments.

### Weaknesses

- The empirical evaluation is relatively limited. Experiments are conducted on a narrow set of benchmarks, and qualitative results appear to select the best outcomes from three hyperparameter settings, which weakens the strength of the empirical conclusions.

- The paper does not compare with mainstream state-of-the-art editing pipelines, such as Qwen-Image-Edit [1] or Kontext [2], which already address the problem of preserving source image structure during semantic editing.

- The only public benchmark used is designed for super-resolution rather than semantic image editing, and commonly used editing benchmarks such as PIE-Bench [3], ImgEdit [4], and GEdit-Bench [5] are not included. This makes it difficult to assess whether the method generalizes to standard editing tasks.

- From a methodological perspective, the core idea of reusing correlated noise to couple source and target generation is already well known in generative modeling. Framing it through coupled SDEs appears to be a relatively natural extension rather than a fundamentally new concept.

[1] Qwen-image technical report

[2] Flux. 1 kontext: Flow matching for in-context image generation and editing in latent space

[3] Pnp inversion: Boosting diffusion-based editing with 3 lines of code

[4] Imgedit: A unified image editing dataset and benchmark

[5] Step1x-edit: A practical framework for general image editing

---

> ### Author Rebuttal · Authors · 2026-03-30
>
> We thank the reviewer for acknowledging the technical soundness of our mathematical formulation and the clarity of our presentation. We address each concern below.
>
> ---
>
> ## Weakness 1:
>
> We agree that a larger-scale evaluation would further strengthen the paper. That said, we note that the scale of our evaluation is consistent with (and in several cases exceeds) that of prior training-free editing methods in this domain. The table below summarizes evaluation dataset sizes from representative training-free editing works:
>
>
> | Method | Dataset Size |
> | --- | --- |
> | Prompt-to-Prompt (Hertz et al., 2023) | < 100, Qualitative results only|
> | SEGA (Brack et al., 2023) | 250 |
> | FlowEdit (Kulikov et al., 2024) | 250 |
> | Fireflow (Deng et al., 2024) | 700 |
> | **Sync-SDE (Ours)** | 306 (our new constructed dataset) + 250 (div2k, in supplementary) = 556  |
>
> ---
>
> ## Weakness 2 & Question 1:
>
> We agree that comparing with strong baselines is important, and we appreciate the suggestion. However, there is a fundamental methodological distinction: [1] and [2] are instruction-following models **pre-trained** on large-scale, expensive (source image, target image, editing instruction) triplets, whereas our approach is entirely training-free, addressing semantic editing from a different perspective grounded in coupled SDEs and optimal transport. Directly comparing these two fundamentally different paradigms, **training-free vs. heavily supervised**, conflates their research contributions, just as the existence of large supervised models does not invalidate zero-shot or unsupervised approaches.
>
> That said, we will include qualitative comparisons against [1] and [2] in the revised manuscript to provide additional context for practitioners.
>
> ---
>
> ## Weakness 3 & Question 2:
>
> We agree that evaluation on established benchmarks is valuable, and we have carefully considered each suggested one:
>
> - **PIE-Bench [3]:** Given its age and prevalence (released in 2023), there is a non-trivial risk that it has **leaked into the training data** of foundation models such as Flux, compromising evaluation validity. In contrast, our dataset is constructed exclusively from images uploaded **after** the release of Flux.1, ensuring it is free from such training data contamination.
> - **ImgEdit [4]:** This benchmark contains *synthetic* (source, target) image pairs, which introduces its own evaluation biases.
> - **ImgEdit [4] and GEdit-Bench [5], format incompatibility:** Both benchmarks use the format [source image, editing instruction, target image], whereas our method requires [source prompt, source image, target prompt]. Converting between formats via LLM-generated prompts is unreliable: prompt quality varies across LLMs, significant human intervention is needed, and derivation errors confound evaluation. An accurate, large-scale conversion is infeasible within our current capacity and would likely introduce more noise than it resolves.
>
> We will cite and discuss [3], [4], and [5] in the revised paper to contextualize our evaluation choices, and note that designing a dedicated benchmark for prompt-guided (rather than instruction-guided) editing remains an open problem. If time allows, we will run experiments on [3] and include them in the appendix.
>
> ---
>
> ## Weakness 4 & Question 4:
>
> We appreciate the reviewer's perspective, and we agree that at a surface level the mechanism may appear similar to prior approaches that reuse correlated noise to couple source and target generation. However, we believe the core technical distinction is significant:
>
> Prior methods reuse **free Gaussian noise**, either by sharing a common noise sample across generation processes or by deterministically inverting a source image via DDIM and replaying the trajectory. Our method, in contrast, operates through a fundamentally different mechanism:
>
> > Source image → construct a **structured reverse Brownian motion** (which does not naturally exist and must be explicitly derived) → synchronize the target sampling process with respect to this structured reverse Brownian motion.
> >
>
> We believe it constitutes a meaningful contribution beyond existing heuristics.
>
> ---
>
> ## Question 3:
>
> We appreciate the reviewer raising this concern, as transparency in evaluation is important to us. We would like to point out that our paper **already provides** comprehensive qualitative results across all hyperparameter settings in the appendix. Specifically, Figures 11, 12, and 13 present side-by-side qualitative comparisons of Sync-SDE against all baselines (FireFlow, FlowEdit, RF-Edit, RF-Inv, and SDEdit) under **all three hyperparameter configurations** from Table 3, with pixel-wise difference maps for each setting. We will add a more prominent pointer to these appendix figures in the revised main text to ensure this transparency is immediately visible to readers.

---

> > ### Author Rebuttal · Reviewer_fdn3 · 2026-04-03
> >
> > I appreciate the clarifications on novelty and the transparency of hyperparameter results in the appendix. However, the lack of concrete experiments on established benchmarks and comparisons with trained SOTA (even qualitative) remains unaddressed in practice—only promised. I maintain my current score.

---

> > > ### Author Response · Authors · 2026-04-07
> > >
> > > Thank you for your constructive feedback and for your patience while we ran these additional experiments. All of the following changes are already integrated in the paper:
> > >
> > > # Scaled Evaluation
> > >
> > > We present the quantitative experimental results on PIE-Bench [3]:
> > >
> > > [PIE-Bench Results](https://anonymous.4open.science/r/random-3580/pie_plot.png)
> > >
> > > The setup is the same as the experiments in the main paper: we run experiments on three hyperparameter settings as recommended in their papers or codebase, and we plot the L1 distance vs. CLIP score and LPIPS vs. CLIP score trade-offs. From the plot, we can see that Sync-SDE still yields the most desirable output on PIE-Bench.
> > >
> > > # Quantitative comparisons against FluxKontext and Qwen-Edit
> > >
> > > To evaluate FluxKontext and Qwen-Edit on our benchmark of 306 source-target prompt pairs, we generate editing instructions from each (source prompt, target prompt) pair using Claude-Sonnet-4.6, which we found to produce the most reliable and semantically faithful instructions among the large language models we tested; we then manually polished the outputs to correct occasional ambiguities. We adopt guidance scales of 3.5 for FluxKontext and 4.0 for Qwen-Edit as suggested in their own papers, and sweep over {2.5, 3.5, 5.0} and {3.0, 4.0, 5.0} respectively to trace the trade-off between prompt adherence and source fidelity. As shown in the figure below, Sync-SDE achieves a more favorable trade-off than both FluxKontext and Qwen-Edit. Moreover, combining Sync-SDE with these instruction-based models further improves source similarity while maintaining competitive CLIP scores.
> > >
> > > [Trade-off Plot](https://anonymous.4open.science/r/random-3580/trained_plot.png)
> > >
> > > # Qualitative comparisons against FluxKontext and Qwen-Edit
> > >
> > > The figure below provides a qualitative comparison of Sync-SDE, FluxKontext, Qwen-Edit, FluxKontext + Sync-SDE, and Qwen-Edit + Sync-SDE using their default hyperparameter choices:
> > >
> > > [Qualitative Comparison](https://anonymous.4open.science/r/random-3580/trained_qual_comp.png)
> > >
> > > FluxKontext and Qwen-Edit often introduce a minor displacement of objects (like a camera shake) without altering their relative positions. When Sync-SDE is combined with either instruction-based model, this displacement is substantially reduced.

---

### Official Review · Reviewer_CgH1 · 2026-03-13

**Soundness:** 4
**Presentation:** 2
**Significance:** 4
**Originality:** 4
**Overall Recommendation:** 5
**Confidence:** 3

**Summary:**

The author present a novel way for semantic editing with diffusion model. The author come up with a way to pin down the set of noise used in reverse diffusion process, such that if they use this noise sequence, the reverse diffusion process lands exactly on the "source image." To edit the image, you replace the score function in the reverse diffusion process with the new conditioning prompt. The author shows very good editing quality with state of the art diffusion model, on par or better than other image editing model.

**Compliance With Llm Reviewing Policy:**

Affirmed.

**Final Justification:**

I read author's rebuttal and all other review, I still recommend acceptance of the paper.

**Key Questions For Authors:**

Has the author thought about how their method fit in a larger context: like using diffusion model for posterior sampling? I am curious because the method provided by the author seems very general. I never thought about controlling the sequence of noise in the reverse diffusion process. To me think is the main contribution and it's super cool. Semantic editing just felt like one application. Do the author thought of any other application? Or could this method be used to understand diffusion model better?

**Limitations:**

Yes

**Strengths And Weaknesses:**

Strength:
1. Very cool idea.
2. I think the idea of controlling the noise sequence in the reverse diffusion process is novel (I never seen it). And the connection to classical theory in physics is also cool.
3. The application to semantic editing is neat. The semantic editing result also looks very good.


Weakness:
The only weakness I can think of is that the presentation is a little to difficult for me to understand. It took me a coupled of read to understand the intuition of why we want to manipulate the noise in the reverse diffusion process and how the author is doing it. I personally would prefer a high level idea how on the coupled dynamics works before the author state the key equation 1.

---

> ### Author Rebuttal · Authors · 2026-03-30
>
> We sincerely thank the reviewer for the enthusiastic and encouraging feedback, and for recognizing the novelty and its connection to classical theory. We are glad the idea resonated with you. We address your points below.
>
> ---
>
> **1. Presentation & Intuition (Weakness)**
>
> We agree that the paper would benefit from a gentler on-ramp before the formal machinery. In the revision, we will add a high-level, intuitive explanation of the coupled dynamics and pathwise correspondence before introducing Equation 1.
>
> ---
>
> **2. Broader Context & Future Applications (Question)**
>
> We greatly appreciate this question as it touches on what we believe is the most exciting long-term direction of this work. We agree that semantic editing is just one application; the core contribution, constructing and controlling structured Brownian path in the reverse diffusion process, is indeed a general-purpose mechanism.
>
> Under the lens of generative models, our approach provides a principled way to **inject controlled correlations** (both positive and negative) between generated samples. For example:
>
> - **Positive correlation (merging):** With proper design of the coupling, one could generate a sample that is consistent with a target conditioning $c_{tar}$ while visually resembling a *blend* of two reference data points $A$ and $B$.
> - **Negative correlation (avoidance):** Conversely, one could generate a sample that stays close to data point $A$ while being pushed *away* from data point $B$.
> - **Posterior sampling:** As the reviewer suggests, the framework naturally connects to posterior sampling with diffusion models, where one wants to generate samples conditioned on observations while maintaining specific structural relationships.
>
> These are all compelling directions for future work. We are happy to carry out preliminary experiments demonstrating some of these capabilities if time allows during the discussion period, and we will expand the discussion of broader applications in the revised manuscript.

---

> > ### Author Rebuttal · Reviewer_CgH1 · 2026-04-02
> >
> > The author answer my question on future direction and presentation style. I will keep my score because I already recommend acceptance.

---

> > > ### Author Response · Authors · 2026-04-08
> > >
> > > We sincerely thank the reviewer for their time and insights! We are glad the response addressed your concerns!

---

### Official Review · Reviewer_5tuV · 2026-03-13

**Soundness:** 3
**Presentation:** 2
**Significance:** 2
**Originality:** 3
**Overall Recommendation:** 4
**Confidence:** 3

**Summary:**

The paper introduces sync-SDE, a training-free and optimization-free method for semantic image editing using pretrained text-to-image models. The core idea is to couple the forward SDE of a source image with the reverse-time SDE of the target image by sharing the exact same backward Brownian motion path. The authors provide a theoretical interpretation of this approach, framing synchronous coupling as a greedy choice for optimal bicausal Monge transport under a local quadratic cost. Experiments conducted exclusively on the Flux.1 [dev] model claim to show improved trade-offs between prompt adherence (CLIP) and structure preservation (L1, LPIPS) compared to existing baselines like FlowEdit and RF-Inv. An alternative method, resampling-ODE, is also proposed in the appendix for better stability.

**Compliance With Llm Reviewing Policy:**

Affirmed.

**Final Justification:**

I appreciate the authors' rebuttal and will maintain my positive rating.

**Key Questions For Authors:**

1. Cross-Model Generalization: Can the authors provide qualitative and quantitative results using other popular diffusion models (e.g., SDXL)? Does the greedy optimal transport assumption hold equally well in standard Latent Diffusion Models as it does in Rectified Flow models like Flux?

2. Structural Editing Limitations: How can sync-SDE be adapted to allow for structural changes (e.g., adding a new object that alters the layout, or completely removing an object)? Is the strict coupling of the Brownian path a fundamental bottleneck for non-rigid edits?

3.Human Evaluation: Could the authors provide a human preference study comparing sync-SDE with FlowEdit and RF-Inv to validate the trade-off curves presented in Figure 2 and Figure 6?

4.Sensitivity to Start Time ($t\_0$): The paper fixes $t\_0 = 1/7$ for main experiments but shows in Figure 16 that the edit strength is highly sensitive to this parameter. How should a user determine the optimal $t\_0$ for different types of prompts in a zero-shot setting?

**Limitations:**

The authors have correctly identified the "over-preservation" limitation in the appendix, noting that the method fails when radical layout changes or object removals are required (e.g., the potato/salt example). However, the lack of broader model testing (only Flux is used) and the absence of human evaluations are critical missing pieces in the current manuscript.

**Strengths And Weaknesses:**

Strengths:

(1) Elegant Theoretical Framing: The mathematical interpretation of reverse-time SDE coupling through the lens of optimal bicausal Monge transport is novel and provides a solid theoretical foundation for noise-sharing techniques in diffusion/flow models.

(2) Efficiency: As a training-free and test-time-optimization-free method, sync-SDE is highly efficient, requiring only standard SDE integration steps.

(3) Strong Texture Preservation: For edits requiring strict layout and texture preservation (e.g., changing material properties or colors), the method yields visually impressive results with minimal background distortion.

Weaknesses:

(1) Severely Limited Empirical Scope (Single Model): The method is proposed as a general SDE-based editing framework, yet it is evaluated exclusively on a single model: Flux.1 [dev]. Given the diverse latent spaces of modern generative models, it is mandatory to validate the framework on other architectures (e.g., Stable Diffusion 1.5, SDXL, or SD3) to prove its generalizability.

(2) "Over-preservation" and Limited Editing Capability: As honestly acknowledged by the authors in Appendix D.6, sync-SDE suffers from an "over-preservation" issue. It struggles with non-rigid edits or object removals. For instance, the model attempts to replace salt with potatoes simply to fill the geometric space, rather than genuinely following the semantic intent of removal. This severely restricts the method's utility to basic "texture/color swapping" rather than true semantic editing.

(3) Lack of Human Evaluation: Image editing is inherently subjective. Relying solely on automated metrics like CLIP score, L1, and LPIPS is insufficient. A method that aggressively preserves the layout will naturally score well on L1/LPIPS, but this does not mean the edit aligns with human intent. The absence of a user study (A/B testing) makes the quantitative claims less convincing.

(4) Disjointed Presentation of resampling-ODE: resampling-ODE is introduced late in the Appendix, yet it is heavily featured in the main quantitative results (Figure 6). If this variant is crucial for stability, it should be integrated into the main methodological discussion rather than treated as an afterthought.

---

> ### Author Rebuttal · Authors · 2026-03-30
>
> We thank the reviewer for recognizing our method's theoretical elegance and efficiency. We address each point below.
>
> **1. Cross-Model Generalization (Weakness1, Question1)**
>
> We agree that cross-model validation is important. We will include results with other models to demonstrate generalization. We note, however, that comprehensive baseline comparisons may be limited, as many baselines only offer official Flux implementations.
>
> **2. Structural Editing & "Over-preservation" (Weakness2, Question2)**
>
> We appreciate the reviewer's careful analysis. The "over-preservation" phenomenon is directly tied to the optimal transport (OT) nature of Sync-SDE. To keep the transportation cost low, the method naturally reuses existing structures (moving less probability mass).
>
> However, structural editing is not fundamentally bottlenecked. In fact, the ability of complete removal is **already** demonstrated in Figure 3 and Figure 7. To further overcome ``over-preservation'', the most straightforward approach is to use a smaller $t_0$ and larger guidance to increase the editing budget. On the other hand, our method could also work with masks or as inpaintings to alter the image structure with extra accuracy.
>
> **3. Human Evaluation (Weakness3, Question3)**
>
> We agree that human evaluation is valuable. We'd like to note, however, that human evaluation is not standard for recent state-of-the-art training-free semantic editing papers. Among the best-performing training-free methods (FireFlow, FlowEdit, RF-Edit, RF-Inv, SDEdit), only RF-Edit provides human evaluation.
>
> Due to a recent institutional move, we are actively assessing our budget for a user study and will include one if financially feasible.
>
> **4. Presentation of Resampling-ODE (Weakness4)**
>
> We appreciate this feedback. The resampling-ODE variant was placed in the appendix solely due to the 8-page submission limit. We will integrate it into the main text using the extra camera-ready page.
>
> **5. Sensitivity to Start Time $t_0$ (Question4)**
>
> We agree that practical guidance would be helpful. We will add this heuristic: use a smaller $t_0$ for large structural edits, and a larger $t_0$ for fine-grained edits.

---

> > ### Author Rebuttal · Reviewer_5tuV · 2026-04-02
> >
> > Thank you for the rebuttal; my concerns are addressed and I maintain my positive score.

---

> > > ### Author Response · Authors · 2026-04-08
> > >
> > > We sincerely thank the reviewer for their time and insights! We are glad the response addressed your concerns!

---

### Decision · Program_Chairs · 2026-04-30

**Decision:**

Accept (regular)

**Comment:**

The paper received uniformly positive evaluations (1 Accept, 3 Weak Accepts). Reviewers acknowledged the theoretical formulation, efficiency, and texture preservation capabilities of the proposed method. Initial concerns centered on the limited empirical scope, the lack of comparisons with trained SOTA models, and the absence of a user study.

In the rebuttal, the authors successfully mitigated the majority of these critiques by presenting new PIE-Bench evaluations and additional baseline comparisons. Specially, the leaning negative reviewer #fdn3 raised the score to positive. The reviewers agree that despite the pending human evaluation, the theoretical insights and enhanced empirical evidence warrant publication. Having reviewed the paper and the discussion thread, the AC aligns with this positive consensus and recommends acceptance, urging the authors to incorporate the promised revisions into the final version.